# Adaptive Mixture of Disentangled Experts for Dynamic Graph Out-of-Distribution Generalization

**Haibo Chen[1], Xin Wang[1,2]\*, Guanheng Chen[1], Yuan Meng[1,2], Haoyang Li[1],**
**Yang Yao[1], Zeyang Zhang[1], Zhiqiang Zhang[3], Jun Zhou[3], Ling Feng[1], Wenwu Zhu[1,2]\***
[1]Department of Computer Science and Technology, Tsinghua University
[2]Beijing National Research Center for Information Science and Technology
[3]Ant Group
{chb24,chengh24,yaoyang21,zy-zhang20}@mails.tsinghua.edu.cn
{xin_wang, fengling, wwzhu}@tsinghua.edu.cn, lihy218@gmail.com
lingyao.zzq@antfin.com, jun.zhoujun@antgroup.com

## Abstract

Dynamic graph out-of-distribution (OOD) generalization has drawn an increasing amount of attention in the research community, given its wide applicability in real-world scenarios. Existing methods typically employ a fixed-architecture design to extract invariant patterns. However, there may exist evolving distribution shifts in dynamic graphs, leading to suboptimal performance of fixed-architecture designs. To address this issue, we propose a novel adaptive-architecture design to handle evolving distribution shifts over time, to the best of our knowledge, for the first time. The proposed adaptive-architecture design introduces an adaptive mixture of architecture experts to capture invariant patterns under evolving distribution shifts, which imposes three challenges: 1) How to detect and characterize evolving distribution shifts to inform architectural decisions; 2) How to dynamically route different expert architectures to handle varying distribution characteristics; 3) How to ensure that the adaptive mixture of experts effectively discovers invariant patterns. To solve these challenges, we propose a novel **Ada**ptive **Mix**ture of Disentangled Experts (**AdaMix**) model to adaptively route architecture experts to varying distribution shifts and jointly learn spatio-temporal invariant patterns. Specifically, we propose a spatio-temporal distribution detector to infer evolving distribution shifts by jointly leveraging historical and current information. Building upon this, we develop a prototype-guided mixture of disentangled experts that adaptively routes experts with disentangled factors to different distribution shifts. Finally, we design a distribution-aware intervention mechanism that discovers invariant patterns based on expert selection of nodes. Extensive experiments on both synthetic and real-world datasets demonstrate that our proposed **AdaMix** model significantly outperforms state-of-the-art baselines. Codes will be available at https://github.com/haibo12/AdaMix.

## 1 Introduction

Dynamic graph out-of-distribution (OOD) generalization (Zhang et al., 2022; 2023a; Yuan et al., 2023; Yang et al., 2024) aims to tackle distribution shifts and ensure effective generalization for dynamic graphs, whose structures and features evolve over time (Li et al., 2019; You et al., 2019; Wu et al., 2020). Existing methods for dynamic graph OOD generalization typically attempt to extract invariant patterns, i.e., structures and features whose predictive abilities remain stable across shifts. For example, DIDA (Zhang et al., 2022) employs a disentangled spatio-temporal graph attention network to encode node trajectories into invariant and variant representations, and then applies random interventions on the variant part to force predictions to rely on the invariant patterns.

---

\*Corresponding Authors.

However, existing methods typically rely on fixed-architecture designs to extract invariant patterns, overlooking that distribution shifts in dynamic graphs are continuously evolving and may require adaptive architectures over time to extract optimal invariant patterns. For instance, in academic collaboration networks, the distribution of research topics may evolve with certain regularities. Typical phenomena include the growth in the number of publications (expanding graph size), the increasing density of citation relationships (rising node degrees), and the diversification of research fields (increasing feature diversity). This example illustrates a broader phenomenon where the characteristics of distribution shifts themselves evolve, exemplifying common distribution shifts in graphs (Gui et al., 2022; Li et al., 2025a). Furthermore, the evolution of these distribution shifts may require changes in architectural requirements over time, as the required architectures are inherently shaped by the underlying data distributions (Niu et al., 2021; Wu et al., 2024).

In this paper, we propose a novel adaptive-architecture design to address evolving distribution shifts in dynamic graphs, leveraging a mixture of experts (MoE) that dynamically adjusts the model architecture over time to capture invariant patterns more effectively. This design to handle evolving distribution shifts remains largely unexplored in the literature, and is highly non-trivial, presenting several key challenges: 1) How to detect and characterize evolving distribution shifts to inform architectural decisions? 2) How to dynamically route different expert architectures to handle varying distribution characteristics? 3) How to ensure that the adaptive mixture of experts effectively discovers invariant patterns?

To address these challenges, we propose a novel **Ada**ptive **Mix**ture of Disentangled Experts (**AdaMix**) method to adaptively route expert networks to different distribution shifts for jointly learning spatio-temporal invariant patterns. Specifically, we propose a spatio-temporal distribution detector to infer evolving distribution shifts based on historical and current information, which includes a memory vector for storing historical distribution information. Then, we develop a prototype-guided mixture of disentangled experts that adaptively routes experts to varying distribution shifts. Each expert is associated with a disentangled prototype that captures a distinct factor of variation. Finally, we design a distribution-aware intervention mechanism that encourages nodes to be intervened upon by others from different distributions, leveraging expert-based interventions to discover invariant patterns. Extensive experiments on real-world and synthetic datasets demonstrate the effectiveness of our proposed method, outperforming state-of-the-art baselines. The contributions of this paper are summarized as follows:

- We propose a novel adaptive-architecture design—Adaptive Mixture of Disentangled Experts (**AdaMix**)—to handle evolving distribution shifts in dynamic graphs, where the adaptive-architecture is defined relative to the underlying data distribution. To the best of our knowledge, this is the first work to address dynamic graph distribution shifts from an architectural perspective.
- We observe that different timestamps under evolving distribution shifts may require distinct architecture designs, and further provide a theoretical analysis demonstrating the advantages of adaptive architectures over fixed ones in such cases.
- We propose three key components to address adaptive mixture of experts for dynamic graph OOD generalization: i) spatio-temporal distribution detector; ii) prototype-guided disentangled experts; and iii) distribution-aware intervention mechanism.
- We conduct extensive experiments on real-world and synthetic datasets, demonstrating the effectiveness of our proposed method, which outperforms state-of-the-art baselines.

## 2 PROBLEM FORMULATION AND NOTATIONS

In this section, we present the fundamental concepts and notations used throughout the paper, focusing on dynamic graphs and distribution shifts within them. Random variables are denoted using **bold** letters (e.g., $\mathbf{G}$), while their realizations are denoted using *italic* letters (e.g., $\mathcal{G}$).

**Dynamic Graphs.** We denote a dynamic graph as $\mathcal{G} = \{\mathcal{G}^t\}_{t=1}^T$, where $T$ denotes the total number of timestamps. Each snapshot $\mathcal{G}^t = (\mathcal{V}^t, \mathcal{E}^t)$ corresponds to the graph at time $t$, with node set $\mathcal{V}^t$ and edge set $\mathcal{E}^t$. For simplicity, a snapshot can also be expressed as $\mathcal{G}^t = (\mathbf{X}^t, \mathbf{A}^t)$, where $\mathbf{X}^t$ is the node feature matrix and $\mathbf{A}^t$ is the adjacency matrix. The prediction task on dynamic graphs aims to leverage historical snapshots to make future predictions, *i.e.*, $p(\mathbf{Y}^t|\mathbf{G}^{1:t})$, where $\mathbf{G}^{1:t} = \{\mathbf{G}^1, \ldots, \mathbf{G}^t\}$ represents the graph trajectory up to time $t$, and $\mathbf{Y}^t$ denotes the target labels

(e.g., node properties or future links) at time $t + 1$. Following (Zhang et al., 2022), the distribution of the entire trajectory can be factorized into ego-graph trajectories: $p(\mathbf{Y}^t \mid \mathbf{G}^{1:t}) = \prod_v p(\mathbf{y}_v^t \mid \mathbf{G}_v^{1:t})$.

**Distribution Shifts in Dynamic Graphs.** The standard learning objective is to optimize a predictor under empirical risk minimization (ERM): $\min_\theta \; \mathbb{E}_{(y_v^t, \mathcal{G}_v^{1:t}) \sim p_{tr}(\mathbf{y}_v^t, \mathbf{G}_v^{1:t})} \mathcal{L}(f_\theta(\mathcal{G}_v^{1:t}), y_v^t)$, where $f_\theta$ is a parameterized dynamic graph neural network. However, under distribution shifts, the predictor trained on the training distribution $p_{tr}$ may not generalize to the test distribution $p_{te}$, since $p_{tr}(\mathbf{Y}^t, \mathbf{G}^{1:t}) \neq p_{te}(\mathbf{Y}^t, \mathbf{G}^{1:t})$. Following the OOD generalization literature (Arjovsky et al., 2019; Wu et al., 2022b; Gagnon-Audet et al., 2022; Zhang et al., 2022), we adopt the following assumption regarding invariant and variant patterns under distribution shifts in dynamic graphs:

**Assumption 1.** *For a given task, suppose there exists a predictor $f(\cdot)$ such that, for any distribution, each sample $(\mathcal{G}_v^{1:t}, y_v^t)$ can be decomposed into two parts: an invariant pattern $P_I^t(v)$ and a variant pattern $P_V^t(v)$. These patterns are required to satisfy: (1) the invariant pattern alone is sufficient for prediction, i.e., $y_v^t = f(P_I^t(v)) + \epsilon$, where $\epsilon$ denotes noise; (2) the invariant pattern can be obtained by excluding the variant pattern from the observed trajectory, i.e., $P_I^t(v) = \mathcal{G}_v^{1:t} \setminus P_V^t(v)$; (3) the effect of the variant pattern on the label is fully shielded by the invariant pattern, i.e., $\mathbf{y}_v^t \perp \mathbf{P}_V^t(v) \mid \mathbf{P}_I^t(v)$.*

## 3 MOTIVATION

In this section, we illustrate the motivation of our proposed method. We begin by introducing the phenomenon of evolving distribution shifts in dynamic graphs, and then discuss the limitations of existing methods from an architectural perspective.

**Evolving Distribution Shifts.** In real-world dynamic graphs, the underlying data distribution evolves continuously over time, *i.e.*, $p(\mathbf{G}^t) \neq p(\mathbf{G}^{t'})$ for $t \neq t'$. To empirically validate this phenomenon, we analyze several dynamic graph datasets. As illustrated in Figure 1, on the Collab academic collaboration network, key graph statistics such as the number of nodes and the average node degree exhibit a continuous upward trend over time. This evolution reflects typical forms of graph distribution shifts (Gui et al., 2022; Li et al., 2025a). Importantly, the presence of some trends implies that historical trajectories contain valuable signals for inferring the current distribution. Furthermore, because the graph distribution continuously evolves, the joint distribution of labels

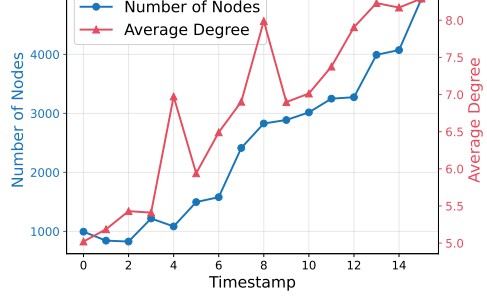

Figure 1: Visualizations of the number of nodes and average degree in each graph snapshot.

and historical graph data also changes accordingly, *i.e.*, $p(\mathbf{Y}^t, \mathbf{G}^{1:t}) \neq p(\mathbf{Y}^{t'}, \mathbf{G}^{1:t'})$. Additional analyses on other datasets are provided in Appendix B.3.

**Architectures Impact.** Existing methods for dynamic graphs typically rely on fixed architectures to extract invariant patterns. However, the optimal architecture may be inherently tied to the underlying data distribution (Niu et al., 2021; Wu et al., 2024). When the data distribution continuously evolves, as is the case in dynamic graphs, a single fixed architecture may become suboptimal over time. We hypothesize that different timestamps under evolving distribution shifts may require distinct architecture designs. To validate this hypothesis, we evaluate two GNN architectures GAT (Veličković et al., 2017) and GATv2 (Brody et al., 2021), built upon a dynamic graph OOD method SILD (Zhang et al., 2023a). Figure 2 presents the performance of each architecture at various timestamps on the Collab dataset. As

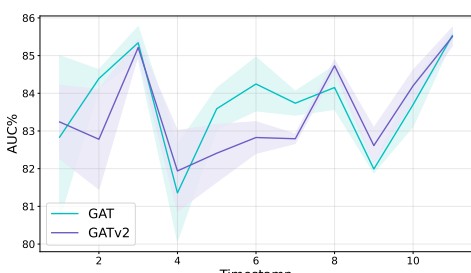

Figure 2: The solid lines indicate the average AUC across timestamps, with the shaded region representing the standard deviation.

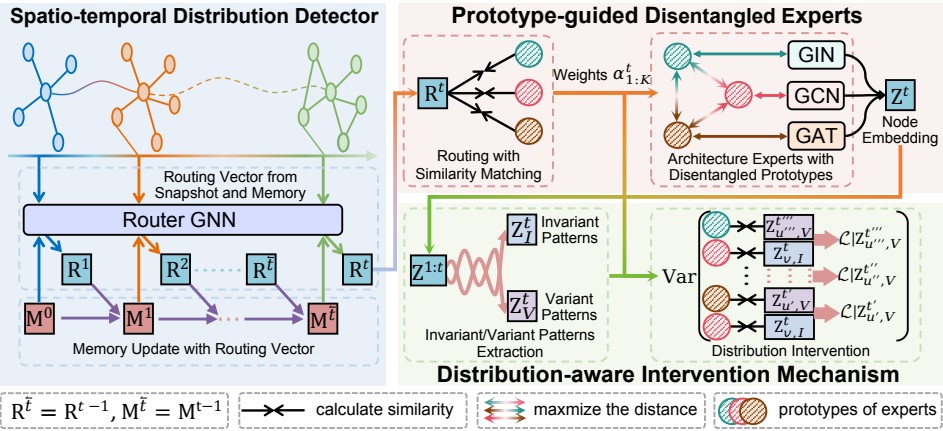

Figure 3: Overview framework of **AdaMix**. Given a snapshot at time $t$, the spatio-temporal distribution detector employs Router GNN to infer the current distribution shift based on the current snapshot and historical information stored in the memory vector, producing a routing vector. In the prototype-guided disentangled experts module, the routing vector is used to compute the weights of different experts by measuring the similarity between the routing vector and the experts' prototypes. The experts' prototypes are simultaneously disentangled to ensure minimal similarity, and the expert outputs are combined according to their weights to obtain the Mixture-of-Experts (MoE) node embeddings. Finally, in the distribution-aware intervention mechanism, all MoE node embeddings prior to time $t$ are decoupled into invariant and variant patterns for the current time $t$, and nodes are selected for intervention training based on the expert weights.

can be seen, GAT and GATv2 alternately achieve superior performance over different time periods, supporting that different timestamps may require distinct architectures. We further provide a theoretical discussion showing that, under the invariance constraint $\mathbf{y}_v^t \perp \mathbf{P}_V^t(v) \mid \mathbf{P}_I^t(v)$ specified in Assumption 1, when different timestamps require distinct architecture designs, adaptive architectures can capture invariant and variant patterns more effectively than any fixed architecture. We formalize this in the following proposition, with the proof deferred to Appendix C.2.

**Proposition 1.** *Under the invariance constraint $\mathbf{y}_v^t \perp \mathbf{P}_V^t(v) \mid \mathbf{P}_I^t(v)$, if there exist two timestamps, $t_1$ and $t_2$, for which the optimal architectures differ when tasked with discovering invariant patterns at $t_1$ and $t_2$, then an adaptive architecture can capture invariant patterns $\mathbf{P}_I^t(v)$ and variant patterns $\mathbf{P}_V^t(v)$ more effectively than a fixed architecture.*

## 4 METHOD

In this section, we propose Adaptive Mixture of Disentangled Experts (**AdaMix**) for dynamic graph OOD generalization. The method comprises three key components: prototype-guided disentangled experts, a spatio-temporal distribution detector, and a distribution-aware intervention mechanism. The overall framework of **AdaMix** is depicted in Figure 3.

### 4.1 PROTOTYPE-GUIDED DISENTANGLED EXPERTS

To route experts adaptively under evolving distribution shifts, it is crucial that each expert specializes in a distinct factor of variation in the data distribution, thereby ensuring alignment with the underlying distribution. However, in standard MoE frameworks, experts operate independently and lack explicit relational modeling, failing to encourage that each expert corresponds to a meaningful or disentangled factor of variation. To address this, we propose prototype-guided disentangled experts, which associate each expert with a corresponding prototype. These prototypes are mutually disentangled, capturing distinct factors of variation, and act as anchors to guide the routing process.

**Disentangled Experts.** We adopt $K$ independent GNN architectures as experts, such as GAT (Veličković et al., 2017) and GCN (Kipf & Welling, 2016), denoted by $\{\text{GNN}_k\}_{k=1}^K$. Each

expert encodes the input graph independently, generating node embeddings:

$$\mathbf{H}_k^t = \text{GNN}_k(\mathbf{X}^t, \mathbf{A}^t), \tag{1}$$

where $\mathbf{H}_k^t = \{\mathbf{h}_{v,k}^t \mid v \in \mathcal{V}\} \in \mathbb{R}^{|\mathcal{V}| \times d_h}$ denotes the node representation matrix produced by expert $k$ at time $t$, and $\mathbf{X}^t$ and $\mathbf{A}^t$ are the node feature matrix and adjacency matrix at time $t$, respectively, and $d_h$ is the hidden dimension. We utilize a set of learnable prototypes $\{\mathbf{p}_k\}_{k=1}^K$ for $K$ experts, where each prototype $\mathbf{p}_k \in \mathbb{R}^{d_h}$ represents a distinct factor of variation. During training, we encourage each expert to specialize in capturing information associated with its corresponding prototype. To this end, we introduce a similarity loss that promotes disentanglement among prototypes:

$$\mathcal{L}_{\text{dis}} = \sum_{k=1}^K \sum_{k' \neq k} \frac{\mathbf{p}_k \cdot \mathbf{p}_{k'}}{\|\mathbf{p}_k\|_2 \|\mathbf{p}_{k'}\|_2}. \tag{2}$$

Minimizing this loss enforces mutual dissimilarity among prototypes, thereby fostering disentanglement across experts and encouraging each to develop a distinct area of specialization.

**Prototype-guided Routing.** Let $\mathbf{r}_v^t$ denote the routing vector for node $v$ at time $t$, which captures the current distribution (introduced in the next section). To route experts in alignment with the underlying distribution, we calculate the similarity between the routing vector $\mathbf{r}_v^t$ and the prototypes $\{\mathbf{p}_k\}_{k=1}^K$ associated with each expert. The resulting similarity scores are then transformed into expert weights $\alpha_{v,k}^t$ via a softmax:

$$\alpha_{v,k}^t = \frac{\exp(\hat{\alpha}_{v,k}^t)}{\sum_{k'=1}^K \exp(\hat{\alpha}_{v,k'}^t)}, \quad \hat{\alpha}_{v,k}^t = \frac{\mathbf{r}_v^t \cdot \mathbf{p}_k}{\|\mathbf{p}_k\|_2}. \tag{3}$$

When the routing vector $\mathbf{r}_v^t$ is more similar to the prototype of a particular expert, that expert is assigned a higher weight. The outputs of all experts are then aggregated to obtain the node embedding $\mathbf{z}_v^t$ for each node $v$ at time $t$:

$$\mathbf{z}_v^t = \sum_{k=1}^K \alpha_{v,k}^t \mathbf{h}_{v,k}^t. \tag{4}$$

MoE node embeddings $\mathbf{Z} = \{\mathbf{z}_v^t \mid v \in \mathcal{V}, \ t = 1, \dots, T\} \in \mathbb{R}^{T \times |\mathcal{V}| \times d_h}$ are subsequently processed to extract both invariant and variant patterns.

## 4.2 SPATIO-TEMPORAL DISTRIBUTION DETECTOR.

To infer specific distribution shifts within the context of evolving distribution shifts, we propose a spatio-temporal distribution detector that leverages both historical and current information. Specifically, our goal is to capture node-level distribution by jointly considering the current ego-graph and historical distributional information.

**Snapshot Graph Trajectories Modeling.** We adopt a $\text{GNN}_r$ to learn a node-level routing embedding $\mathbf{r}_v^t$ for each node $v$ at time $t$ from its ego-graph trajectory $\mathcal{G}_v^t = (\mathcal{V}_v^t, \mathcal{E}_v^t)$, which encodes the structural and feature information of the current snapshot. Formally,

$$\mathbf{r}_v^t = \text{GNN}_r(\mathcal{V}_v^t, \mathcal{E}_v^t), \quad \mathcal{V}_v^t = \{v\} \cup \mathcal{N}^t(v), \quad \mathcal{E}_v^t = \{(u,v) \in \mathcal{E}^t \mid u \in \mathcal{V}^t\}, \tag{5}$$

where $\mathcal{N}^t(v) = u \mid (u,v) \in \mathcal{E}^t$. High-order structural information can be captured by stacking multiple GNN layers or employing advanced architectures such as GAT (Veličković et al., 2017). This produces the node-level routing embedding matrix $\mathbf{R}^t = \{\mathbf{r}_v^t \mid v \in \mathcal{V}^t\} \in \mathbb{R}^{|\mathcal{V}| \times d_h}$ for all nodes in the snapshot at time $t$.

**Memory-augmented Vector.** To infer the distribution from historical information, we utilize a memory bank $\mathbf{M} = \{\mathbf{m}_v \mid v \in \mathcal{V}\} \in \mathbb{R}^{|\mathcal{V}| \times d_h}$, which stores historical distributional information for all nodes. For each node $v$, we denote the memory vector $\mathbf{m}_v^t \in \mathbb{R}^{d_h}$ that accumulates its historical information up to time $t$. At each step, the routing vector $\mathbf{r}_v^t$ is generated by $\text{GNN}_r$, which integrates the initial node embedding $\mathbf{x}_v^t$ with the previous memory vector $\mathbf{m}_v^{t-1}$:

$$\mathbf{r}_v^t = \text{GNN}_r(\tilde{\mathbf{x}}_v^t, \mathbf{A}^t), \quad \tilde{\mathbf{x}}_v^t = \text{Linear}([\mathbf{x}_v^t \| \mathbf{m}_v^{t-1}]), \tag{6}$$

where $\tilde{\mathbf{x}}_v^t \in \mathbb{R}^{d_x}$ is the combined feature of node $v$ at time $t$, $d_x$ denotes the dimension of the initial node features, $\|$ denotes the node-wise concatenation operation, and $\text{Linear}(\cdot)$ is a linear transformation to project the concatenated feature to the input dimension of $\text{GNN}_r$. We then obtain the routing weights $\boldsymbol{\alpha}_v^t$ and MoE node embeddings $\mathbf{z}_v^t$ for each node $v$ using the prototype-guided routing mechanism described in Eq. 3 and Eq. 4, respectively. Finally, we update the memory bank through a gate-controlled mechanism:

$$\mathbf{m}_v^t = \mathbf{g}_v^t \odot \mathbf{z}_v^t + \left(\mathbf{1} - \mathbf{g}_v^t\right) \odot \mathbf{m}_v^{t-1}, \mathbf{g}_v^t = \sigma\left(\text{Linear}_{\text{gate}}\left([\mathbf{z}_v^t \parallel \mathbf{m}_v^{t-1}]\right)\right) \in [0,1]^{d_h}, \quad (7)$$

where $\odot$ denotes element-wise multiplication, $\mathbf{g}_v^t \in [0,1]^{d_h}$ is a gate vector that controls the update rate of the memory, and $\sigma(\cdot)$ is the sigmoid function. We set the initial memory vector $\mathbf{m}_v^0$ to a zero vector. This mechanism allows the memory to adaptively incorporate new information while retaining relevant historical context, thereby allowing the routing vectors $\mathbf{r}_v^t$ to infer current distribution shifts from both present and past information.

## 4.3 Distribution-Aware Intervention Mechanism

Previous dynamic graph OOD methods typically rely on randomly sampling variant patterns to replace those of other nodes to discover invariant patterns. However, such interventions may be inefficient when some nodes are intervened upon by others from the same distribution. To address this, we leverage the expert weights from previous steps to distinguish nodes from different distributions better, and we apply interventions using nodes sampled from distinct distributions.

**Invariant and Variant Patterns.** We first extract the invariant and variant patterns based on MoE node embeddings $\mathbf{Z} = \{\mathbf{z}_v^t\} \in \mathbb{R}^{T \times |\mathcal{V}| \times d_h}$ in Eq 4. To account for distribution shifts that may be unobservable in the time domain but become apparent in the spectral domain (Zhang et al., 2023a), we apply a Fast Fourier transform (FFT) to project $\mathbf{Z}$ into the spectral domain. Let $\text{Re}(\mathbf{Z}) = \{\text{Re}(\mathbf{z}_v^t)\} \in \mathbb{R}^{T \times |\mathcal{V}| \times d_h}$ and $\text{Im}(\mathbf{Z}) = \{\text{Im}(\mathbf{z}_v^t)\} \in \mathbb{R}^{T \times |\mathcal{V}| \times d_h}$ denote the real and imaginary parts of the transformed embeddings, respectively. We then derive disentangled invariant and variant spectrum masks $\mathbf{m}_I$ and $\mathbf{m}_V$ as follows:

$$\mathbf{m}_I = \sigma\left(\frac{\mathbf{m}}{\tau}\right), \quad \mathbf{m}_V = \sigma\left(-\frac{\mathbf{m}}{\tau}\right), \quad \mathbf{m} = \text{MLP}(\text{Re}(\mathbf{Z}) \| \text{Im}(\mathbf{Z})), \quad (8)$$

where $\sigma(\cdot)$ denotes the sigmoid function, $\tau$ is a temperature hyperparameter, and $\text{MLP}(\cdot)$ is a multi-layer perceptron. Finally, the invariant and variant patterns $\mathbf{Z}_I$ and $\mathbf{Z}_V$ are obtained as follows:

$$\mathbf{Z}_I = \text{IFFT}\left(\text{Re}(\mathbf{Z}) \odot \mathbf{m}_I + i\,\text{Im}(\mathbf{Z}) \odot \mathbf{m}_I\right), \ \mathbf{Z}_V = \text{IFFT}\left(\text{Re}(\mathbf{Z}) \odot \mathbf{m}_V + i\,\text{Im}(\mathbf{Z}) \odot \mathbf{m}_V\right), \quad (9)$$

where $\text{IFFT}(\cdot)$ denotes the inverse Fast Fourier transform, and $i$ is the imaginary unit. $\mathbf{Z}_I = \{\mathbf{z}_{v,I}^t\} \in \mathbb{R}^{T \times |\mathcal{V}| \times d_h}$ and $\mathbf{Z}_V = \{\mathbf{z}_{v,V}^t\} \in \mathbb{R}^{T \times |\mathcal{V}| \times d_h}$ represent the invariant and variant patterns for all nodes across all timestamps, respectively.

**Distribution-Aware Intervention.** Since experts are assigned to nodes according to their underlying distributions, a large difference in dominant experts between two nodes strongly suggests that they follow different distributions. We first identify the dominant expert $e_v^t$ for each node based on the routing weights $\alpha_{v,k}^t$ in Eq. 3:

$$e_v^t = \arg\max_k \alpha_{v,k}^t. \quad (10)$$

To ensure that nodes are intervened upon by others from distinct distributions, we intervene on nodes by sampling other nodes with different dominant experts to replace their variant patterns. Specifically, at each time step $t$, we randomly sample a set of nodes $u$ from the invariant patterns $\mathbf{Z}_I$ (e.g., $e_u^{t'}$), and then replace the variant pattern of node $v$ if its dominant expert at time $t$ differs from that of $u$ at time $t'(t' \leq t)$. Consequently, the invariance loss is defined as follows:

$$\mathcal{L}_{\text{inv}} = \text{Var}(\mathcal{L}|\mathbf{z}_{u,V}^{t'} : \mathbf{z}_{u,V}^{t'} \in \mathbf{Z}_V), \quad (11)$$

$$s.t. \quad \mathcal{L}|\mathbf{z}_{u,V}^{t'} = \sum_{v=1}^{N} l\left(f_I(\mathbf{z}_{v,I}^t) \cdot (\sigma(\mathbf{z}_{u,V}^{t'}) \cdot \mathbf{1}_{e_v^t \neq e_u^{t'}} + 1 \cdot \mathbf{1}_{e_v^t = e_u^{t'}}), \mathbf{y}_v^t\right), \quad (12)$$

where $f_I(\cdot)$ is a classifier based on invariant patterns, $l(\cdot, \cdot)$ is the cross-entropy loss, and $\mathbf{1}_{a_u^t \neq a_v^t}$ is an indicator function that equals 1 if $a_u^t \neq a_v^t$ and 0 otherwise. In this way, we ensure that nodes are intervened upon by others from different distributions, thereby enhancing the effectiveness of the intervention mechanism. Then, we calculate the final loss as follows:

$$\mathcal{L} = \mathcal{L}_\mathrm{I} + \lambda \underbrace{\mathcal{L}_\mathrm{inv}}_{\uparrow \text{Eq. 12}} + \alpha \underbrace{\mathcal{L}_\mathrm{dis}}_{\uparrow \text{Eq. 2}}, \tag{13}$$

where $\mathcal{L}_\mathrm{I}$ is the empirical risk based on invariant patterns, and $\lambda$ and $\alpha$ are hyperparameters that balance the three loss terms. The overall training procedure of **AdaMix** is summarized in Algorithm 1.

## 5 EXPERIMENTS

In this section, we conduct extensive experiments to demonstrate that our proposed method effectively handles distribution shifts on dynamic graphs through an adaptive MoE framework. Additional details regarding experimental settings and supplementary results are provided in the Appendix B.

### 5.1 EXPERIMENTAL SETUP

**Datasets.** We evaluate our method on three real-world dynamic graph datasets that exhibit evolving distribution shifts. For the task of link prediction, we use two datasets: Collab (Tang et al., 2012), an academic collaboration network spanning papers published from 1990 to 2006, and Yelp (Sankar et al., 2020), which contains customer reviews of businesses over a 24-month period. For both datasets, the data is partitioned such that the test set contains different fields from those used in training, thereby simulating a real-world distribution shift. For node classification, we use the Aminer dataset (Tang et al., 2008; Sinha et al., 2015), a citation network covering papers published from 2001 to 2015. In addition, we employ synthetic datasets (Zhang et al., 2023a) with different levels of distribution shifts(0.4, 0.6, 0.8) to further validate the effectiveness of our method. Figures in Appendix B.3 illustrate the evolving distribution shifts across these datasets. Additional details about the datasets are provided in Appendix B.1.

**Baselines.** We compare our proposed **AdaMix** with three categories of baselines: (1) representative dynamic GNNs, including GCRN (Seo et al., 2018), EGCN (Pareja et al., 2020), and DySAT (Sankar et al., 2020); (2) general OOD generalization methods, including IRM (Arjovsky et al., 2019), GroupDRO (Sagawa et al., 2019), and V-REx (Krueger et al., 2021); (3) static graph MoE methods, including GMoE (Wang et al., 2023) and GraphMETRO (Wu et al., 2024); and (4) dynamic graph OOD generalization methods, including DIDA (Zhang et al., 2022), EAGLE (Yuan et al., 2023) and SILD (Zhang et al., 2023a). To ensure that the performance gains are not merely due to introducing specialized architectures, we replace the original architecture in SILD (Zhang et al., 2023a) with the same architecture experts used in our method for comparison, including GCN (Kipf & Welling, 2016), GAT (Veličković et al., 2017), GIN (Xu et al., 2018), and GATv2 (Brody et al., 2021). More details about the baselines are provided in Appendix B.2.

### 5.2 MAIN RESULTS

**Real-world Datasets.** Following Zhang et al. (2023a), we evaluate the performance of different methods on real-world datasets with distribution shifts split, details of which are provided in Appendix B.1. Table 1 presents the results of different methods on real-world datasets. From Table 1, we have the following observations: (1) Dynamic graph OOD methods generally achieve better performance than both dynamic GNNs and general OOD methods, highlighting the importance of incorporating temporal information when addressing distribution shifts in dynamic graphs. However, results on the Aminer dataset show that dynamic graph methods cannot guarantee optimal performance at all time periods. For instance, when SILD employs GATv2 as its backbone, it achieves the best performance on Aminer15 but underperforms GAT on Aminer17. This suggests that distribution shifts may differ across time, necessitating different architectures to handle them effectively. (2) Our proposed **AdaMix** achieves superior or competitive performance on most datasets, often surpassing existing baselines. These results highlight its effectiveness in handling distribution shifts in dynamic graphs, with the adaptive MoE framework enabling better adaptation to evolving distribution shifts.

Table 1: Performance of different methods on real-world link prediction and node classification datasets. The best results are highlighted in bold, and the second-best are underlined. For the Aminer dataset, the year indicates the test split, *e.g.*, 'Aminer15' refers to the average test accuracy in 2015.

| Task Dataset | Link Prediction (AUC%) | | Node Classification (ACC%) | | | Avg. |
|---|---|---|---|---|---|---|
| | Collab | Yelp | Aminer15 | Aminer16 | Aminer17 | |
| GCRN | $69.72_{\pm0.45}$ | $54.68_{\pm7.59}$ | $47.96_{\pm1.12}$ | $51.33_{\pm0.62}$ | $42.93_{\pm0.71}$ | 57.27 |
| EGCN | $76.15_{\pm0.91}$ | $53.82_{\pm2.06}$ | $44.14_{\pm1.12}$ | $46.28_{\pm1.84}$ | $37.71_{\pm1.84}$ | 57.56 |
| DySAT | $76.59_{\pm0.20}$ | $66.09_{\pm1.42}$ | $48.41_{\pm0.81}$ | $49.76_{\pm0.96}$ | $42.39_{\pm0.62}$ | 63.18 |
| IRM | $75.42_{\pm0.87}$ | $56.02_{\pm16.08}$ | $48.44_{\pm0.13}$ | $50.18_{\pm0.73}$ | $42.40_{\pm0.27}$ | 59.48 |
| VREx | $76.24_{\pm0.77}$ | $66.41_{\pm1.87}$ | $48.70_{\pm0.73}$ | $49.24_{\pm0.27}$ | $42.59_{\pm0.37}$ | 63.16 |
| GroupDRO | $76.33_{\pm0.29}$ | $66.97_{\pm0.61}$ | $48.73_{\pm0.61}$ | $49.74_{\pm0.26}$ | $42.80_{\pm0.36}$ | 63.46 |
| GMoE | $56.45_{\pm0.56}$ | $72.53_{\pm15.14}$ | $49.17_{\pm1.54}$ | $50.89_{\pm1.61}$ | $43.14_{\pm0.61}$ | 58.90 |
| GraphMETRO | $57.92_{\pm0.11}$ | $45.66_{\pm10.59}$ | $50.05_{\pm0.17}$ | $52.12_{\pm1.96}$ | $42.29_{\pm1.91}$ | 50.58 |
| DIDA | $81.87_{\pm0.40}$ | $75.92_{\pm0.90}$ | $50.34_{\pm0.81}$ | $51.43_{\pm0.27}$ | $44.69_{\pm0.06}$ | 68.87 |
| EAGLE | $\underline{84.41}_{\pm0.87}$ | $77.26_{\pm0.74}$ | $51.48_{\pm0.45}$ | $\mathbf{54.87}_{\pm0.31}$ | $\underline{45.97}_{\pm0.23}$ | 70.81 |
| SILD | $84.09_{\pm0.16}$ | $78.65_{\pm2.22}$ | $52.35_{\pm1.04}$ | $54.11_{\pm0.62}$ | $45.54_{\pm1.19}$ | $\underline{71.14}$ |
| SILD-GCN | $79.53_{\pm0.70}$ | $43.74_{\pm0.24}$ | $50.54_{\pm0.87}$ | $53.47_{\pm0.60}$ | $41.64_{\pm2.96}$ | 57.27 |
| SILD-GAT | $83.82_{\pm0.25}$ | $50.18_{\pm0.75}$ | $51.68_{\pm1.81}$ | $53.93_{\pm1.89}$ | $44.87_{\pm1.42}$ | 61.39 |
| SILD-GIN | $75.18_{\pm0.42}$ | $\underline{81.55}_{\pm0.67}$ | $49.04_{\pm1.92}$ | $51.15_{\pm1.63}$ | $23.68_{\pm17.22}$ | 66.01 |
| SILD-GATv2 | $83.97_{\pm0.12}$ | $47.84_{\pm1.96}$ | $\underline{52.70}_{\pm1.54}$ | $54.15_{\pm0.93}$ | $43.35_{\pm3.14}$ | 60.63 |
| **AdaMix** | $\mathbf{84.85}_{\pm0.39}$ | $\mathbf{82.65}_{\pm0.87}$ | $\mathbf{52.95}_{\pm0.70}$ | $\underline{54.58}_{\pm0.20}$ | $\mathbf{46.50}_{\pm0.63}$ | **72.95** |

Table 2: Performance of different methods on synthetic link prediction and node classification datasets. The best results are highlighted in bold, and the second-best are underlined. A larger shift indicates a higher level of distribution shift.

| Dataset Shift | Link-Synthetic (AUC%) | | | Node-Synthetic (ACC%) | | | Avg. |
|---|---|---|---|---|---|---|---|
| | 0.4 | 0.6 | 0.8 | 0.4 | 0.6 | 0.8 | |
| GCRN | $72.57_{\pm0.72}$ | $72.29_{\pm0.47}$ | $67.26_{\pm0.22}$ | $27.19_{\pm2.18}$ | $25.95_{\pm0.80}$ | $29.26_{\pm0.69}$ | 49.09 |
| EGCN | $69.00_{\pm0.53}$ | $62.70_{\pm1.14}$ | $60.13_{\pm0.89}$ | $24.01_{\pm2.29}$ | $22.75_{\pm0.96}$ | $24.98_{\pm1.32}$ | 43.93 |
| DySAT | $70.24_{\pm1.26}$ | $64.01_{\pm0.19}$ | $62.19_{\pm0.39}$ | $40.95_{\pm2.89}$ | $37.94_{\pm1.01}$ | $30.90_{\pm1.97}$ | 51.04 |
| IRM | $69.40_{\pm0.09}$ | $63.97_{\pm0.37}$ | $62.66_{\pm0.33}$ | $33.23_{\pm4.70}$ | $30.29_{\pm1.71}$ | $29.43_{\pm1.38}$ | 48.16 |
| VREx | $70.44_{\pm1.08}$ | $63.99_{\pm0.21}$ | $62.21_{\pm0.40}$ | $41.78_{\pm1.30}$ | $38.11_{\pm2.81}$ | $29.56_{\pm0.44}$ | 51.02 |
| GroupDRO | $70.30_{\pm1.23}$ | $64.05_{\pm0.21}$ | $62.13_{\pm0.35}$ | $41.35_{\pm2.19}$ | $35.74_{\pm3.93}$ | $31.03_{\pm1.24}$ | 50.77 |
| GMoE | $55.39_{\pm1.92}$ | $54.97_{\pm4.94}$ | $56.30_{\pm2.35}$ | $\underline{83.33}_{\pm1.04}$ | $80.83_{\pm0.06}$ | $72.08_{\pm1.31}$ | 67.15 |
| GraphMETRO | $59.53_{\pm0.08}$ | $59.28_{\pm0.09}$ | $58.72_{\pm0.12}$ | $\underline{75.82}_{\pm4.35}$ | $78.19_{\pm3.53}$ | $\underline{75.25}_{\pm3.82}$ | 67.80 |
| DIDA | $85.20_{\pm0.84}$ | $82.89_{\pm0.23}$ | $72.59_{\pm3.31}$ | $43.33_{\pm7.74}$ | $39.48_{\pm7.93}$ | $28.14_{\pm3.07}$ | 58.60 |
| EAGLE | $\underline{88.32}_{\pm0.61}$ | $\underline{87.29}_{\pm0.71}$ | $\underline{82.30}_{\pm0.75}$ | $47.03_{\pm0.10}$ | $35.84_{\pm1.05}$ | $28.50_{\pm0.16}$ | 61.55 |
| SILD | $85.95_{\pm0.18}$ | $84.69_{\pm1.18}$ | $78.01_{\pm0.71}$ | $43.62_{\pm2.74}$ | $39.78_{\pm3.56}$ | $38.64_{\pm2.76}$ | 61.78 |
| SILD-GCN | $69.43_{\pm0.19}$ | $63.16_{\pm0.12}$ | $60.64_{\pm0.08}$ | $78.59_{\pm1.00}$ | $73.21_{\pm2.62}$ | $65.93_{\pm3.51}$ | $\underline{68.49}$ |
| SILD-GAT | $85.97_{\pm0.15}$ | $84.69_{\pm1.11}$ | $78.01_{\pm0.61}$ | $43.15_{\pm4.21}$ | $40.15_{\pm1.95}$ | $38.51_{\pm2.09}$ | 61.75 |
| SILD-GIN | $60.73_{\pm1.01}$ | $58.99_{\pm1.31}$ | $55.22_{\pm0.80}$ | $77.89_{\pm2.12}$ | $74.65_{\pm3.38}$ | $63.36_{\pm4.09}$ | 65.14 |
| SILD-GATv2 | $86.19_{\pm0.43}$ | $83.82_{\pm0.14}$ | $68.43_{\pm0.59}$ | $41.48_{\pm0.85}$ | $40.18_{\pm2.30}$ | $38.08_{\pm1.03}$ | 59.70 |
| **AdaMix** | $\mathbf{90.21}_{\pm0.13}$ | $\mathbf{89.64}_{\pm0.26}$ | $\mathbf{88.86}_{\pm0.13}$ | $\mathbf{83.63}_{\pm1.60}$ | $\mathbf{81.50}_{\pm0.38}$ | $\mathbf{76.19}_{\pm0.82}$ | **85.00** |

**Synthetic Datasets.** Table 2 reports the results on six synthetic datasets. (1) We observe that **AdaMix** outperforms most baselines across the datasets, indicating that its adaptive architecture effectively captures invariant patterns under varying levels of distribution shift. (2) As the degree of distribution shift increases, the performance of all baselines degrades significantly. In contrast, **AdaMix** shows a smaller performance drop, further demonstrating its strong ability to handle distribution shifts. (3) The performance of SILD varies significantly when using different GNN architectures as experts, indicating that no single architecture is optimal for all distribution shifts. In contrast, **AdaMix** consistently achieves strong performance by adaptively selecting the most suitable architectures for each node at each time step.

## 5.3 ABLATION STUDY

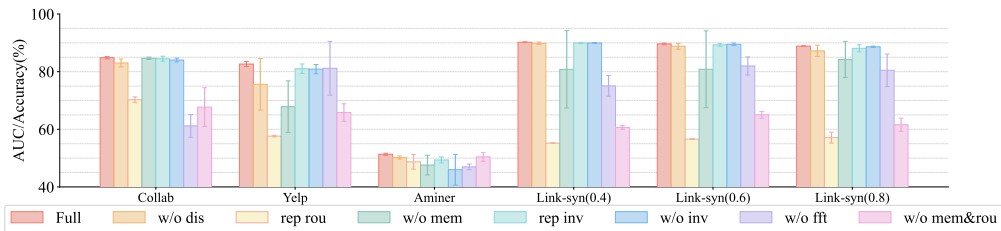

Figure 4: Comparisons of different ablated versions of **AdaMix** on real-world and synthetic datasets.

To verify the effectiveness of each designed component in **AdaMix** , we compare different ablated versions on each dataset: 1) **w/o dis**: we remove the disentanglement loss $\mathcal{L}_{\text{dis}}$ in Eq. 2 by setting $\alpha = 0$ in Eq. 13; 2) **rep rou**: we replace the prototype-guided routing mechanism with a simple linear router that directly maps the routing embeddings to expert weights; 3) **w/o mem**: we remove the memory-augmented mechanism by setting the memory vector always to a zero vector; 4) **rep inv**: we replace the invariance loss in Eq. 12 with a random intervention mechanism that randomly samples nodes from the invariant patterns to replace the variant patterns; 5) **w/o inv**: we remove the invariance loss $\mathcal{L}_{\text{inv}}$ in Eq. 12 by setting $\lambda = 0$ in Eq. 13. 6) **w/o fft**: we remove the FFT and IFFT operations in Eqs. 9 when extracting invariant and variant patterns. 7) **w/o mem&rou**: we remove both the memory-augmented mechanism and replace the prototype-guided routing mechanism with a deeper linear router.

The results are shown in Figure 4. We have the following observations: i) The ablated versions **w/o dis** and **rep rou** exhibit a significant drop and unstable performance on some datasets, indicating that disentangled prototypes help the router better distinguish different distributions, thereby selecting the appropriate experts. ii) The ablated version **w/o mem** and **w/o mem&rou** leads to a noticeable performance decrease, indicating that leveraging the historical distribution information stored in memory vectors enables better inference of the current distribution. Moreover, disentangled prototypes allow the router to distinguish more effectively between different distributions. iii) The ablated version **rep inv** and **w/o inv** yield suboptimal performance, demonstrating the effectiveness of the expert-based interventions in discovering invariant patterns. iv) The ablated version **w/o fft**,which relies solely on time-domain information, shows noticeable declines compared to the full model. This demonstrates that spectral-domain invariant pattern modeling effectively captures distribution shifts that may be unobservable in the time domain but become evident in the spectral domain.

## 5.4 TIME COMPLEXITY ANALYSIS

Let $|\mathcal{V}|$, $|\mathcal{E}|$, and $T$ denote the number of nodes, edges, and time steps, respectively. We denote the dimensions of input features and hidden embeddings by $d_x$ and $d_h$, respectively, and let $|\mathcal{S}|$ represent the number of intervention times. The time complexity of **AdaMix** mainly consists of the following components: the time complexity of $K$ experts is $\mathcal{O}(KT|\mathcal{E}|d_x + KT|\mathcal{V}|d_xd_h)$; the time complexity of the router GNN is $\mathcal{O}(T|\mathcal{E}|d_h + T|\mathcal{V}|d_h^2)$; the time complexity of computing MoE weights is $\mathcal{O}(T|\mathcal{V}|Kd_h)$, and the time complexity of computing MoE node embeddings is also $\mathcal{O}(T|\mathcal{V}|Kd_h)$. In addition, the time complexity of extracting invariant and variant patterns is $\mathcal{O}(T|\mathcal{V}|d_h \log T)$, and the time complexity of distribution-aware interventions is $\mathcal{O}(|\mathcal{S}|T|\mathcal{V}|d_h)$. Therefore, the overall time complexity of **AdaMix** is: $\mathcal{O}\Big(KT|\mathcal{E}|d_x + KT|\mathcal{V}|d_xd_h + T|\mathcal{E}|d_h +$

$T|\mathcal{V}|d_h^2 + T|\mathcal{V}|Kd_h + T|\mathcal{V}|d_h \log T + |\mathcal{S}|T|\mathcal{V}|d_h\Big)$, which scales linearly with the number of edges and nodes in the dynamic graph, which is comparable to existing dynamic graph OOD generalization methods (Zhang et al., 2023a; Yuan et al., 2023).

## 6 RELATED WORK

**Dynamic Graph Neural Networks.** Dynamic graphs are pervasive in numerous real-world scenarios (Deng et al., 2020; Wang et al., 2021; Cai et al., 2021; Zhang et al., 2023b; 2024c;a; Wang et al.,

2025b), ranging from social interactions and recommendation systems to event prediction (Skarding et al., 2021; Zhu et al., 2022; Chen et al., 2023a). One paradigm employs snapshot-based GNNs to learn node representations at each time step and then applies temporal modules such as recurrent or attention-based models to capture temporal evolution (Yang et al., 2021; Sun et al., 2021; Hajiramezanali et al., 2019; Seo et al., 2018). Another paradigm integrates temporal encoding mechanisms that directly embed temporal information into time-aware representations, which are then processed with GNNs or memory architectures (Cong et al., 2021; Xu et al., 2020). Despite these advances, the impact of distribution shifts on dynamic graphs has received limited attention. Some recent works (Zhang et al., 2022; Yuan et al., 2023; Yang et al., 2024; Tieu et al., 2025) have begun to explore this area. For instance, SILD (Zhang et al., 2023a) proposes a spectral-domain method to disentangle invariant and variant spectral patterns in dynamic graphs, thereby achieving generalization against distribution shifts (especially those that are unobservable in the time domain). However, existing methods typically rely on a single model architecture, which may not be optimal for handling evolving distribution shifts over time.

**Graph Mixture of Experts.** Mixture of experts (MoE) models have recently been applied to graph learning to handle the diverse structures and features inherent to graph data (Hu et al., 2021; Liu et al., 2023; Rumiantsev & Coates, 2024; Han et al., 2024; Yao et al., 2025; Ye et al., 2025). An MoE architecture comprises multiple expert networks specialized for different patterns, along with a gating network that selects or weights their outputs. For instance, GMoE (Wang et al., 2023) proposes that each node dynamically routes to one of several information aggregation experts, each with differing hop sizes, so as to better adapt to local graph structure in large-scale settings. Mowst (Zeng et al., 2023) takes a different perspective, utilizing a weak MLP and a strong GNN expert, with a confidence gate that per-node decides how much to rely on feature-only vs. structure-aware prediction. In the context of OOD generalization, GraphMETRO (Wu et al., 2024) uses a Mixture-of-Experts architecture to decompose complex distribution shifts into multiple components. A gating network infers which shifts affect each graph, and each expert is trained to produce representations invariant to its designated shift. However, these methods focus on static graphs, whereas we propose an adaptive MoE framework for dynamic graphs, routing experts based on historical and current information to discover invariant patterns more effectively under evolving distribution shifts.

**Invariant Representation Learning.** Deep invariant representation learning aims to achieve out-of-distribution generalization by capturing stable relationships between data and tasks, thereby enabling more robust prediction (Arjovsky et al., 2019; Ahuja et al., 2020a; Li et al., 2022c; 2023; Xia et al., 2023; Zhang et al., 2024b; Sun et al., 2024; Wang et al., 2024; Chen et al., 2025; Li et al., 2025b). For instance, DIR (Wu et al., 2022b) discovers causal rationales that remain invariant across different distributions, while suppressing spurious patterns that are unstable. EERM (Wu et al., 2022a) proposes an invariant learning framework that employs adversarially trained graph structure editors to simulate virtual environments, enabling GNNs to extrapolate beyond the single observed environment and thus achieve robust node-level prediction. However, existing invariant representation learning methods adopt fixed model architectures to learn invariant representations, overlooking the fact that dynamic graphs with evolving distribution shifts may require adaptive architectures.

## 7 CONCLUSION

In this paper, we study distribution shifts in dynamic graphs from an architectural perspective. We propose **AdaMix**, a novel adaptive mixture-of-experts framework that dynamically selects the most suitable architecture for each node at every time step based on its inferred distribution. Specifically, **AdaMix** employs a spatio-temporal distribution detector to infer the underlying distribution of each node by leveraging both historical and current information. It then incorporates a prototype-guided disentangled experts module, which ensures that each expert specializes in a distinct factor of variation, thereby enabling effective routing. Finally, a distribution-aware intervention mechanism is introduced to enhance the discovery of invariant patterns by intervening nodes with others from different distributions. Extensive experiments on both real-world and synthetic datasets demonstrate the effectiveness of our proposed method. One limitation of our work is that we mainly focus on node-level tasks, and we leave the exploration of graph-level tasks for future work.

## ACKNOWLEDGEMENTS

This work is supported by the National Key Research and Development Program of China No.2023YFF1205001.

## ETHICS STATEMENT

All authors of this work have adhered to the ICLR Code of Ethics. In preparing this manuscript, we have ensured that no human subjects were directly involved, and all data used are publicly available benchmark datasets. To assist with language clarity and grammatical correctness, a large language model (LLM) was employed for proofreading and text refinement; however, all scientific content, ideas, analyses, and conclusions are solely the work of the authors. We have carefully considered potential biases, fairness, and reproducibility of our methods, and we confirm that our research does not involve applications or insights that could cause harm. All experiments comply with applicable legal and ethical standards in machine learning research.

## REPRODUCIBILITY STATEMENT

We have made every effort to ensure the reproducibility of the results reported in this paper. All datasets used in our experiments are publicly available, with data preprocessing steps detailed in Appendix B.1. The proposed Adaptive Mixture of Disentangled Experts (**AdaMix**) model is described with detailed algorithmic steps in Algorithm 1, hyperparameter settings in Appendix C.1, and experimental environment configurations in Appendix C.2.

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

## LLM USAGE STATEMENT

In this work, we leveraged a large language model (LLM) to assist with checking for grammatical errors and improving the clarity and readability of the manuscript. Specifically, the LLM was used to proofread sentences, suggest stylistic improvements, and ensure that the text adhered to formal academic writing standards. All scientific content, ideas, and results presented in this paper are solely the work of the authors.

## A NOTATIONS AND ALGORITHM

Table 3: The summary of the notations and their descriptions used in this paper.

| Notations | Descriptions |
|---|---|
| $\mathcal{G} = (\mathcal{V}, \mathcal{E})$ | Dynamic graph with node set $\mathcal{V}$ and edge set $\mathcal{E}$ |
| $\mathcal{G}^t = (\mathcal{V}^t, \mathcal{E}^t)$ | Graph snapshot at time $t$ with node set $\mathcal{V}^t$ and edge set $\mathcal{E}^t$ |
| $\mathcal{G}_v^t = (\mathcal{V}_v^t, \mathcal{E}_v^t)$ | Ego-graph trajectory of node $v$ at time $t$ |
| $\mathcal{G}_v^{1:t} = (\mathcal{V}_v^{1:t}, \mathcal{E}_v^{1:t})$ | Historical ego-graph trajectory of node $v$ from time 1 to $t$ |
| $d_x, d_h$ | Dimensions of input features and hidden embeddings |
| $\mathbf{X}^t \in \mathbb{R}^{|\mathcal{V}^t| \times d_x}, \mathbf{A}^t \in \mathbb{R}^{|\mathcal{V}^t| \times |\mathcal{V}^t|}$ | Node feature matrix and adjacency matrix at time $t$ |
| $\mathbf{y}_v^t$ | Label of node $v$ at time $t$ |
| $\mathbf{H}_k^t \in \mathbb{R}^{|\mathcal{V}| \times d_h}$ | Node embedding matrix produced by expert $k$ at time $t$ |
| $\mathbf{p}_k \in \mathbb{R}^{d_h}$ | Learnable prototype for expert $k$ |
| $\mathbf{r}_v^t \in \mathbb{R}^{d_h}$ | Routing vector for node $v$ at time $t$ |
| $\mathbf{m}_v^t \in \mathbb{R}^{d_h}$ | Memory vector for node $v$ at time $t$ |
| $\alpha_{v,k}^t$ | Weight of expert $k$ for node $v$ at time $t$ |
| $\mathbf{m}_I$ and $\mathbf{m}_V$ | Invariant and variant masks |
| $\mathbf{z}_v^t \in \mathbb{R}^{d_h}$ | MoE node embedding for node $v$ at time $t$ |
| $\mathbf{Z}_I, \mathbf{Z}_V \in \mathbb{R}^{T \times |\mathcal{V}| \times d_h}$ | Invariant and variant patterns for all nodes across all timestamps |
| $e_v^t$ | Dominant expert for node $v$ at time $t$ |
| $\mathcal{L}_I, \mathcal{L}_{\text{inv}}, \mathcal{L}_{\text{dis}}$ | Empirical risk based on invariant patterns, invariance loss, and disentanglement loss |
| $\lambda, \alpha$ | Hyperparameters to balance different loss terms |

---

**Algorithm 1** Training pipeline for **AdaMix**

---

**Input:** data $\mathcal{D} = \{(\mathcal{G}_v^{1:t}, y_v^t)\}$, number of experts $K$, hyperparameters $\lambda$ and $\alpha$
**Initialize:** experts $\{\text{GNN}_k\}_{k=1}^K$, prototypes $\{\mathbf{p}_k\}_{k=1}^K$, initial memory bank $\mathbf{M}$, distribution detector $\text{GNN}_r$
**for** each epoch **do**
   Reset memory bank $\mathbf{M}$
   **for** each time step $t = 1$ to $T$ **do**
      **for** each node $v \in \mathcal{V}^t$ **do**
         Obtain routing vector $\mathbf{r}_v^t$ using Eq. 6.
         Calculate expert weights $\alpha_{v,k}^t$ using Eq. 3.
         Obtain MoE node embedding $\mathbf{z}_v^t$ using Eq. 4.
         Update memory vector $\mathbf{m}_v^t$ using Eq. 7.
      **end for**
      Calculate invariant and variant masks $\mathbf{m}_I$ and $\mathbf{m}_V$ using Eq. 8.
      Extract invariant and variant patterns $\mathbf{Z}_I$ and $\mathbf{Z}_V$ using Eq. 9.
      Calculate loss $\mathcal{L}$ using Eq. 13.
   **end for**
**end for**

---

## B EXPERIMENT DETAILS AND ADDITIONAL RESULTS

### B.1 DATASETS DETAILS

We summarize the dataset statistics in Table 4 and describe the dataset details as follows.

Table 4: Dataset statistics

| Dataset | Task | # Nodes | # Edges | # Snapshots | Time Granularity | # Features | Evolving Features |
|---|---|---|---|---|---|---|---|
| Collab | Link | 23,035 | 151,790 | 16 | Year | 32 | No |
| Yelp | Link | 13,095 | 65,735 | 24 | Month | 32 | No |
| Aminer | Node | 43,141 | 851,527 | 17 | Year | 128 | No |
| Link-Synthetic | Link | 151,790 | 18,974 | 16 | - | 64 | Yes |
| Node-Synthetic | Node | 5,000 | 11,252,385 | 100 | - | 4 | No |

**Collab** (Tang et al., 2012) is an academic collaboration dataset comprising 16 graph snapshots of co-authored papers published between 1990 and 2006. Nodes represent authors, and edges denote co-authorship relationships. Each edge is annotated with one of five domain-specific attributes: "Data Mining", "Database", "Medical Informatics", "Theory", and "Visualization". For OOD generalization experiments, we designate "Data Mining" as the shifted attribute. The dataset is chronologically split into 10/1/5 graph snapshots for training, validation, and testing, respectively. The full dataset comprises 23,035 authors and 151,790 co-authorship links in total.

**Yelp** (Sankar et al., 2020) is a business review dataset where nodes represent customers or businesses, and edges denote review behaviors. We utilize data from January 2019 to December 2020 (24 graph snapshots), selecting users and reviews with more than 10 interactions. Node features are extracted using word2vec (Mikolov et al., 2013) from reviews, averaged to form 32-dimensional representations for both users and businesses. The distribution shift arises from the COVID-19 pandemic and differing business categories, including "Pizza", "American (New) Food", "Coffee & Tea", "Sushi Bars", and "Fast Food". We designate "Pizza" as the shifted attribute and use 15/1/8 chronological graph slices for training, validation, and testing, respectively. The dataset comprises 13,095 nodes and 65,375 links in total.

**Aminer** (Tang et al., 2008; Sinha et al., 2015) is a citation network constructed by aggregating data from multiple academic sources, including DBLP, ACM, MAG, and others. The dataset comprises research papers and their citation relationships. For our experiments, we focus on predicting the publication venue of a paper. We select the top 20 venues in the dataset as target categories. We use word2vec (Mikolov et al., 2013) to extract 128-dimensional features from paper abstracts and average to obtain paper features. The distribution shift in this task might be attributed to the significant rise of deep learning research. Therefore, we use papers published between 2001 and 2011 for training, those published between 2012 and 2014 for validation, and papers published from 2015 onwards for testing.

**Link-Synthetic** (Zhang et al., 2022) is a synthetic dataset designed to evaluate OOD generalization under controlled spatio-temporal shifts. It is constructed by augmenting the COLLAB dataset. We generate a synthetic feature set $\mathbf{X}_2^t$ by training embeddings to reconstruct future links $\tilde{\mathbf{A}}^{t+1}$ using a cross-entropy loss $\ell(\mathbf{X}_2^t(\mathbf{X}_2^t)^\top, \tilde{\mathbf{A}}^{t+1})$. This ensures $\mathbf{X}_2^t$ encodes strong, spurious correlations with future link patterns. The input features are $\mathbf{X}^t = [\mathbf{X}_1^t \| \mathbf{X}_2^t]$, where $\mathbf{X}_1^t$ are the original COLLAB features. The intensity of the distribution shift is controlled by a time-varying sampling probability $p(t) = \mathrm{clip}(\bar{p} + \sigma \cos(t), 0, 1)$, where $\bar{p}$ is set to 0.4, 0.6, or 0.8 for training and 0.1 for testing. We preserve the dataset division method of training, validation, and testing time steps of 10, 1, and 5.

**Node-Synthetic** (Zhang et al., 2023a) is designed to simulate distribution shifts in node classification tasks by explicitly modeling frequency components on dynamic graphs that exhibit invariant correlations with labels, while others do not. To construct this dataset, we employ a stochastic block model (SBM) (Holland et al., 1983) to generate links between nodes, where the link probability between nodes depends on their class labels. Specifically, the SBM is parameterized as $\mathrm{SBM}(\mathbf{p}_{\mathrm{in}}, p_{\mathrm{out}})$, with $\mathbf{p}_{\mathrm{in}} \in [0, 1]^{C \times 1}$ denoting the intra-class link probability and $p_{\mathrm{out}}$ representing the inter-class link probability. We set $C = 5$ classes for the node labels. Each node is associated with two types of frequency parameters: $f_{\mathrm{low}} \in \{0.02, 0.04, 0.08, 0.10, 0.12\}$ and $f_{\mathrm{high}} \in \{0.22, 0.24, 0.28, 0.30, 0.32\}$. The correlation between $f_{\mathrm{low}}$ and labels is varied across training (0.4), validation (0.6), and testing (0.8) splits, while $f_{\mathrm{high}}$ maintains a fixed correlation of 1 with labels across all splits. At each time step $t$, the dynamic graph $\mathcal{G}^t$ is constructed by aggregating multiple subgraphs: (1) a random graph $\mathcal{G}_r^t$ generated from Gaussian noise, (2) an invariant graph $\mathcal{G}_I^t = \mathrm{SBM}(\mathbf{p}_{\mathrm{in}}^{\mathrm{high}}(t), p_{\mathrm{out}})$ derived from high-frequency parameters, and (3) a variant graph $\mathcal{G}_V^t = \mathrm{SBM}(\mathbf{p}_{\mathrm{in}}^{\mathrm{low}}(t), p_{\mathrm{out}})$ based on low-frequency parameters. The temporal evolution of these parameters is governed by $\mathbf{p}_{\mathrm{in}}^{\mathrm{low}}(t, f) = S_1 (2 + \cos(2\pi f t))$ and

$\mathbf{p}_{\text{in}}^{\text{high}}(t, f) = S_2 \left(2 + \cos(2\pi f t)\right)$, where $p_{\text{out}}, S_1, S_2$ are set to 1e-3, 1e-2, 5e-3 respectively. Each node is assigned 4-dimensional random features to enhance realism. To ensure generalization under distribution shifts, models must identify and prioritize the invariant graph component ($\mathcal{G}_I^t$) for accurate predictions, as the variant component ($\mathcal{G}_V^t$) exhibits unstable label relationships across training and testing phases. This design enables rigorous evaluation of a model's ability to disentangle and leverage invariant spectral patterns in dynamic graphs.

## B.2 BASELINES DETAILS

We adopt several representative dynamic GNNs and Out-of-Distribution(OOD) generalization methods as our baselines:

- Dynamic GNNs: **GCRN** (Seo et al., 2018) integrates a spatial graph convolutional network (GCN) (Kipf & Welling, 2016) with a temporal gated recurrent unit (GRU) (Cho et al., 2014) to capture both structural and temporal dependencies in dynamic graphs. **EGCN** (Pareja et al., 2020) dynamically evolves GCN parameters over time by incorporating an LSTM (Hochreiter & Schmidhuber, 1997) or GRU (Cho et al., 2014), enabling adaptive modeling of network evolution. **DySAT** (Sankar et al., 2020) employs structural self-attention mechanisms to aggregate neighborhood information at each timestamp and uses temporal self-attention to model dynamic network patterns.

- general OOD generalization methods: **IRM** (Arjovsky et al., 2019) seeks to learn a domain-invariant predictor by minimizing the maximum empirical risk across training domains. **Group-DRO** (Sagawa et al., 2019) prioritizes domains with higher prediction errors during training, reducing worst-case risks across heterogeneous environments. **V-REx** (Krueger et al., 2021) minimizes the variance of empirical risks across training domains to enhance generalization under distributional shifts. Although these methods focus on static graphs, they are adapted here by leveraging the best-performing DGNNs as backbone architectures for dynamic graph tasks.

- static graph MoE methods: **GMoE** (Wang et al., 2023) utilizes a mixture-of-experts architecture where each expert captures information at different hop sizes, allowing dynamic routing based on local graph structures. **GraphMETRO** (Wu et al., 2024) employs a mixture-of-experts framework to decompose complex distribution shifts into multiple components, with each expert learning representations invariant to its designated shift.

- dynamic graph OOD generalization methods: **DIDA** (Zhang et al., 2022) captures invariant and variant patterns by utilizing disentangled attention in the spatial-temporal domain, and conducts a spatial-temporal intervention mechanism to let the model abandon spurious features and turning to utilizing invariant features to make predictions. **EAGLE** (Yuan et al., 2023)uses an EA-DGNN to disentangle multi-channel environments. Then, an ECVAE infers and generates diverse environment samples for fine-grained causal interventions. **SILD** (Zhang et al., 2023a) disentangles the frequency components of node feature trajectories in the spectral domain, and then captures invariant patterns by masking out variant frequency components. **SILD-GCN**, **SILD-GAT**, **SILD-GATv2**, **SILD-GIN** apply the SILD framework using the GCN (Kipf & Welling, 2016),GAT (Veličković et al., 2017), GATv2 (Brody et al., 2021) and GIN (Xu et al., 2018) backbone, respectively. We implement these variants to ensure a fair comparison with our **AdaMix** model.

## B.3 CASE STUDY OF EVOLVING DISTRIBUTION SHIFTS

We visualize the evolving distribution shifts in real-world dynamic graphs in terms of the number of nodes and average degree in Figure 5. The distributions of these key graph statistics change significantly over time, confirming the presence of continuous distribution shifts. While some datasets, such as Collab, exhibit a consistent monotonic trend (e.g., continuous growth), this observation suggests that analyzing historical trends can be crucial for inferring the current graph distribution.

## B.4 CASE STUDY OF ARCHITECTURE IMPACT

To effectively demonstrate that different time periods in a dynamic graph require distinct optimal architectures, we conduct a case study using two GNN architectures: GAT (Veličković et al., 2017)

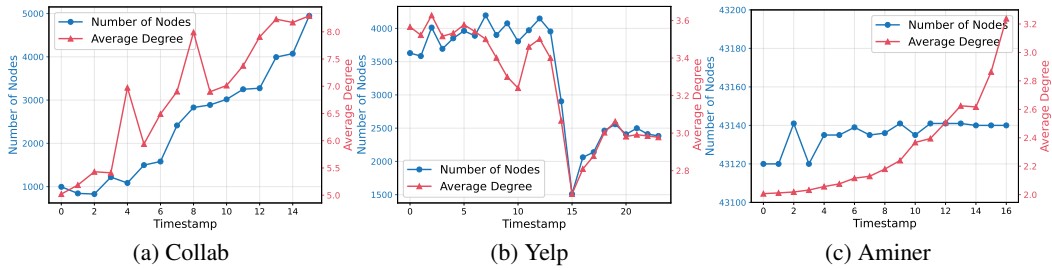

Figure 5: Visualizations of the number of nodes and average degree in each graph snapshot.

and GATv2 (Brody et al., 2021). Built upon the SILD framework (Zhang et al., 2023a), we visualize the timestamp-wise performance of both architectures in Figure 6. Our results reveal that the optimal architecture is not static; GAT outperforms GATv2 at certain timestamps, while GATv2 demonstrates superior performance at others. This finding indicates that no single fixed architecture is sufficient for all time periods, underscoring the critical need for adaptive architectures to handle evolving distribution shifts in dynamic graphs effectively.

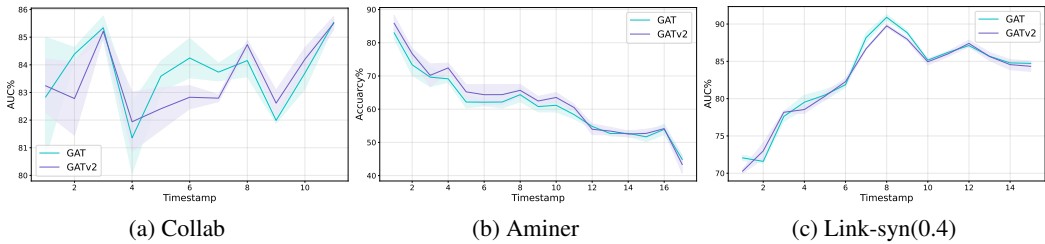

Figure 6: Performance comparison of GAT and GATv2 on the real-world dynamic graphs. The solid lines indicate the average AUC across timestamps, with the shaded region representing the standard deviation.

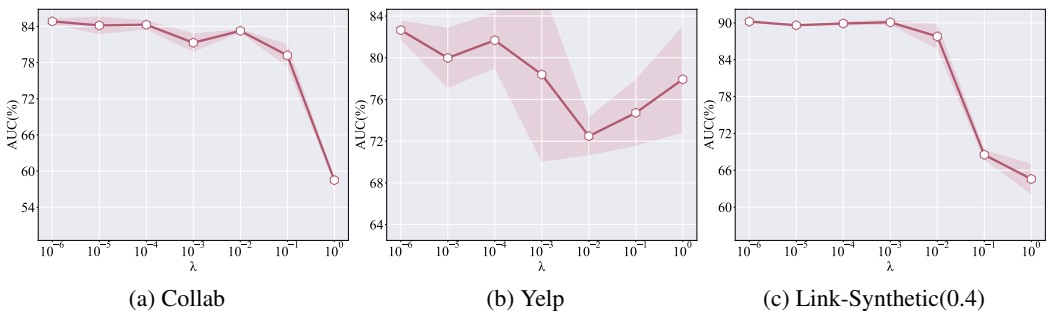

Figure 7: Sensitivity analysis of hyperparameter $\lambda$ on real-world and synthetic datasets. The solid line represents the average AUC (%), with the shaded area showing the standard deviation. The dashed line indicates the average AUC (%) of baseline SILD.

## B.5 HYPERPARAMETERS SENSITIVITY ANALYSIS

We conduct the sensitivity analysis on two key hyperparameters: the weight of the invariance loss, $\lambda$, and the weight of the disentanglement loss, $\beta$. We vary both $\lambda$ and $\beta$ from $\{10^{-6}, 10^{-5}, 10^{-4}, 10^{-3}, 10^{-2}, 10^{-1}, 10^{0}\}$, while keeping all other hyperparameters fixed. The results on both real-world and synthetic datasets are presented in Figure 7 and Figure 8, respectively. The hyperparameter $\lambda$ in Eq. 13 controls the trade-off between minimizing the empirical risk from

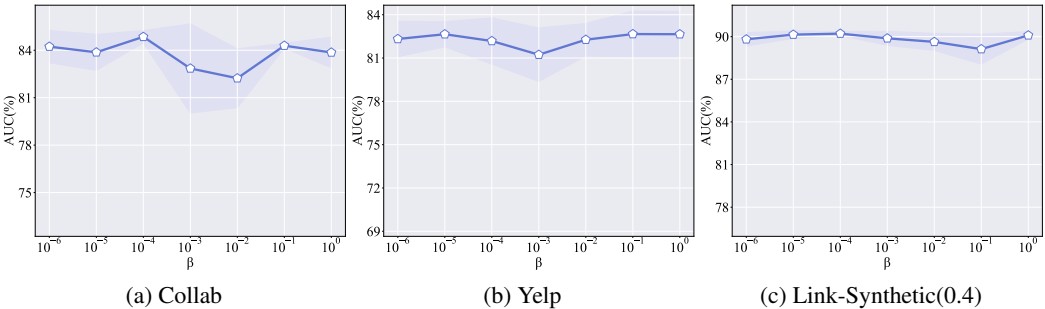

| (a) Collab | (b) Yelp | (c) Link-Synthetic(0.4) |

Figure 8: Sensitivity analysis of hyperparameter $\beta$ on real-world and synthetic datasets. The solid line represents the average AUC (%), with the shaded area showing the standard deviation. The dashed line indicates the average AUC (%) of baseline SILD.

predictions ($L_I$) and enhancing generalization through learning invariant patterns ($L_{\text{inv}}$), as defined in Eq. 12. A large value of $\lambda$ could lead to an over-emphasis on invariance, potentially causing underfitting of the invariant patterns. Similarly, the hyperparameter $\beta$ in Eq. 13 controls the trade-off between $L_I$ and the disentanglement loss ($L_{\text{dis}}$), as defined in Eq. 2. $L_{\text{dis}}$ is crucial for encouraging each expert to learn distinct factors, which is necessary for capturing diverse distribution shifts in dynamic graphs. As shown in Figure 8, our model yields stable performance across a wide range of $\beta$ values, demonstrating that the contribution of the disentanglement loss is robust to its hyperparameter selection.

## B.6 SHOWCASE OF ADAPTIVE ARCHITECTURES

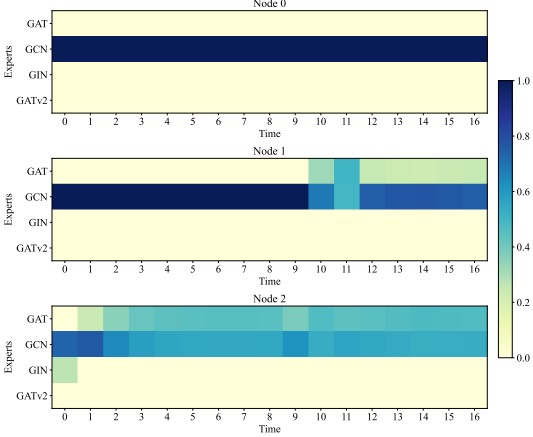

Figure 9: Illustration of the final architectures for Nodes 0–2 from Aminer across different time steps.

As shown in Figure 9, we present the final architectures discovered for Nodes 0–2 in the Aminer dataset. Several observations support our earlier hypotheses: architectures searched for different nodes at the same time step can differ, and architectures for the same node can change over time. Notably, Node 1 exhibits significant architectural changes between early and later stages. In contrast, some nodes maintain consistent architectures, such as Node 0, while Node 2 shows only minor variations, suggesting that the underlying distribution for some nodes may not experience substantial shifts.

## B.7 TRAINING AND INFERENCE TIME

As shown in Table 5, we evaluated the training and inference cost of our approach compared with competitive baselines. The training time is measured on the portion of the "W/O DS" dataset that requires loss computation and backpropagation (i.e., its training split). The inference time is measured on the remaining parts of the dataset, including the validation and test splits of "W/O DS" as well as the entire "W DS" dataset. The results are as follows. The table reports, for each method and each dataset, the average per-epoch training and inference cost (in seconds). All measurements are obtained under the same hardware configuration for a fair comparison. We can observe that environment-modeling methods such as EAGLE incur substantially higher time cost compared to non-environment-modeling approaches. In contrast, our AdaMix introduces only modest overhead relative to SILD.

Table 5: Training and inference time (in seconds) of different methods on various datasets.

| Dataset | EAGLE Train | EAGLE Inf | SILD Train | SILD Inf | AdaMix Train | AdaMix Inf |
|---------|-------------|-----------|------------|----------|--------------|------------|
| Yelp | 6.84 | 0.87 | 0.93 | 0.74 | 0.78 | 0.67 |
| Collab | 14.16 | 3.88 | 0.37 | 0.20 | 0.45 | 0.56 |
| Link-syn (0.4) | 4.96 | 1.23 | 0.35 | 0.36 | 0.47 | 0.71 |
| Link-syn (0.6) | 7.77 | 1.99 | 0.27 | 0.31 | 0.32 | 0.62 |
| Link-syn (0.8) | 11.02 | 2.20 | 0.53 | 0.34 | 0.48 | 0.69 |
| Aminer | 9.22 | 0.18 | 0.31 | 0.27 | 1.03 | 1.15 |
| Node-syn (0.4) | 3.44 | 0.43 | 0.29 | 0.16 | 1.04 | 0.55 |
| Node-syn (0.6) | 3.47 | 0.44 | 0.23 | 0.12 | 1.17 | 0.60 |
| Node-syn (0.8) | 3.48 | 0.44 | 0.22 | 0.12 | 1.14 | 0.60 |

### B.8 PERFORMANCE WITH MORE EXPERTS

To evaluate sensitivity to the number of experts, we add GraphConv (Morris et al., 2019) to the original set of four expert architectures and examined the effect of increasing the number of experts. As shown in Table 6, the results indicate that using five experts still achieves comparable performance.

Table 6: Performance comparison with different numbers of experts.

| Dataset | Collab | Yelp | Link-syn (0.4) | Link-syn (0.6) | Link-syn (0.8) |
|---------|--------|------|----------------|----------------|----------------|
| Four experts | $84.85 \pm 0.39$ | $82.65 \pm 0.87$ | $90.21 \pm 0.13$ | $89.64 \pm 0.26$ | $88.86 \pm 0.13$ |
| Five experts | $85.24 \pm 0.27$ | $83.59 \pm 0.19$ | $88.71 \pm 0.41$ | $89.34 \pm 0.77$ | $88.78 \pm 0.42$ |

## C IMPLEMENTATION DETAILS

### C.1 HYPERPARAMETERS

For all baseline models, we use their official implementations and carefully tune hyperparameters to ensure the best possible performance. For the SILD framework (Zhang et al., 2023a), we replace its original backbone with several widely-used GNN architectures: GCN (Kipf & Welling, 2016), GAT (Veličković et al., 2017), GATv2 (Brody et al., 2021), and GIN (Xu et al., 2018). For our method, we also use these four GNNs as our experts to ensure a fair comparison, maintaining consistent layer and dimension configurations. To ensure a fair comparison, we adopt the same hyperparameter search spaces as the baselines for shared parameters, including the number of attention heads, normalization methods, and dropout rates. For our method's specific hyperparameters, the invariance loss weight $\lambda$ and the disentanglement loss weight $\beta$, we perform a grid search over the set $\{10^{-6}, 10^{-5}, 10^{-4}, 10^{-3}, 10^{-2}, 10^{-1}, 10^0\}$ and empirically select the optimal values for each dataset. We use different learning rates for the expert network and other modules, and we fine-tune both the learning rate and weight decay for each dataset via a grid search on the validation split.

## C.2 Configurations

We conduct all experiments in the following configurations.

- **Operating System**: Ubuntu 24.04.3 LTS
- **CPU**: AMD EPYC 7543 32-Core Processor
- **GPU**: NVIDIA A100-SXM4-40GB and NVIDIA A100-SXM4-80GB
- **Software**: Python 3.9, CUDA 11.7, Pytorch (Paszke et al., 2019) 2.0.1

## D Proof

**Proposition** 1: *Under the invariance constraint $\mathbf{y}_v^t \perp \mathbf{P}_V^t(v) \mid \mathbf{P}_I^t(v)$, if there exist two timestamps, $t_1$ and $t_2$, for which the optimal architectures differ when tasked with discovering invariant patterns at $t_1$ and $t_2$, then an adaptive architecture can capture invariant patterns $\mathbf{P}_I^t(v)$ and variant patterns $\mathbf{P}_V^t(v)$ more effectively than a fixed architecture.*

We provide a proof from the perspective of mutual information. Consider dynamic graphs $\{\mathbf{G}^{1:t}, \mathbf{Y}^t\}_{t=1}^T$, where $\mathbf{G}^{1:t} = (\mathbf{G}^1, \mathbf{G}^2, \ldots, \mathbf{G}^t)$ denotes the sequence of graph snapshots up to time $t$, and $\mathbf{Y}^t$ represents the labels at time $t$. At each timestamp $t$, an encoder architecture $\phi^t$ extracts two invariant patterns and variant patterns $\mathbf{P}_I^t(v)$ and $\mathbf{P}_V^t(v)$:

$$\mathbf{P}_I^t(v) = f_I(\phi^t(\mathbf{G}_v^{1:t})), \quad \mathbf{P}_V^t(v) = f_V(\phi^t(\mathbf{G}_v^{1:t})) \tag{14}$$

We denote the $K$ candidate architectures as $\phi_k$ for $k = 1, 2, \ldots, K$, and define the following two sets:

- **Fixed architectures** $\phi_{\text{fix}} = \{\phi_k^t \mid t = 1, 2, \ldots, T\}$: a single architecture $\phi_k$ is shared across all timestamps $t$. We denote the set containing all $\phi_{\text{fix}}$ as $\Phi_{\text{fix}}$.
- **Adaptive architectures** $\phi_{\text{ada}} = \{\phi_{S(\mathbf{G}^{1:t})}^t \mid t = 1, 2, \ldots, T\}$: the architecture $\phi$ is allowed to vary with $\mathbf{G}_v^{1:t}$, where $S$ is a routing variable that depends on $\mathbf{G}_v^{1:t}$. We denote the set containing all $\phi_{\text{ada}}$ as $\Phi_{\text{ada}}$.

To satisfy the invariance constraint in Assumption 1, we aim to minimize the conditional mutual information $I(\mathbf{P}_V^t(v); \mathbf{y}_v^t \mid \mathbf{P}_I^t(v))$. For each timestamp $t$, we aim to find a $\phi^t$ that achieve :

$$I(\mathbf{P}_V^t(v); \mathbf{y}_v^t \mid \mathbf{P}_I^t(v)) = I\big(f_V(\phi^t(\mathbf{G}_v^{1:t})); \mathbf{y}_v^t \mid f_I(\phi^t(\mathbf{G}_v^{1:t}))\big) = \varepsilon, \tag{15}$$

where $\varepsilon$ is a sufficiently small constant. We then define the constraint-satisfying subsets $\mathcal{F}_{\text{fix}}(\varepsilon)$ and $\mathcal{F}_{\text{ada}}(\varepsilon)$ of $\Phi_{\text{fix}}$ and $\Phi_{\text{ada}}$, respectively, as follows:

$$\mathcal{F}_{\text{fix}}(\varepsilon) = \left\{ \phi_{\text{fix}} \in \Phi_{\text{fix}} \mid I\big(\mathbf{P}_V^t(v); \mathbf{y}_v^t \mid \mathbf{P}_I^t(v)\big) = \varepsilon, \ \forall t \right\} \tag{16}$$

$$\mathcal{F}_{\text{ada}}(\varepsilon) = \left\{ \phi_{\text{ada}} \in \Phi_{\text{fix}} \mid I\big(\mathbf{P}_V^t(v); \mathbf{y}_v^t \mid \mathbf{P}_I^t(v)\big) = \varepsilon, \ \forall t \right\} \tag{17}$$

Clearly, $\Phi_{\text{fix}} \subset \Phi_{\text{ada}}$ (since setting $S$ to a constant recovers a fixed architecture), which implies $\mathcal{F}_{\text{fix}}(\varepsilon) \subset \mathcal{F}_{\text{ada}}(\varepsilon)$. Then we apply the chain rule of mutual information:

$$I\big((\mathbf{P}_I^t(v), \mathbf{P}_V^t(v)); \mathbf{y}_v^t\big) = I(\mathbf{P}_I^t(v); \mathbf{y}_v^t) + I\big(\mathbf{P}_V^t(v); \mathbf{y}_v^t \mid \mathbf{P}_I^t(v)\big) \tag{18}$$

Therefore, under the invariance constraint $I(\mathbf{P}_V^t(v); \mathbf{y}_v^t \mid \mathbf{P}_I^t(v)) = \varepsilon$, maximizing $I((\mathbf{P}_I^t(v), \mathbf{P}_V^t(v)); \mathbf{y}_v^t)$ is equivalent to maximizing $I(\mathbf{P}_I^t(v); \mathbf{y}_v^t)$. Overall, for any $\phi \in \mathcal{F}(\varepsilon)$, we have:

$$\sup_{\mathcal{F}(\varepsilon)} \sum_{t=1}^T I\big((\mathbf{P}_I^t(v), \mathbf{P}_V^t(v)); \mathbf{y}_v^t\big) = \sup_{\mathcal{F}(\varepsilon)} \sum_{t=1}^T \left[ I\big(\mathbf{P}_I^t(v); \mathbf{y}_v^t\big) + I\big(\mathbf{P}_V^t(v); \mathbf{y}_v^t \mid \mathbf{P}_I^t(v)\big) \right] \tag{19}$$

$$= \sup_{\mathcal{F}(\varepsilon)} \sum_{t=1}^T \left[ I\big(\mathbf{P}_I^t(v); \mathbf{y}_v^t\big) + \varepsilon \right] \tag{20}$$

The adaptive architectures can capture both invariant and variant patterns at least as effectively as fixed architectures under the same invariance constraint. Since $\mathcal{F}_{\text{fix}}(\varepsilon) \subset \mathcal{F}_{\text{ada}}(\varepsilon)$, it follows that:

$$\sup_{\mathcal{F}_{\text{fix}}(\varepsilon)} \sum_{t=1}^{T} I\big((\mathbf{P}_I^t(v), \mathbf{P}_V^t(v)); \mathbf{y}_v^t\big) \leq \sup_{\mathcal{F}_{\text{ada}}(\varepsilon)} \sum_{t=1}^{T} I\big((\mathbf{P}_I^t(v), \mathbf{P}_V^t(v)); \mathbf{y}_v^t\big) \tag{21}$$

Similarly, for any subset $\mathcal{T} \subseteq \{1, \ldots, T\}$, we can derive an analogous result:

$$\sup_{\mathcal{F}_{\text{fix}}(\varepsilon)} \sum_{t \in \mathcal{T}} I\big((\mathbf{P}_I^t(v), \mathbf{P}_V^t(v)); \mathbf{y}_v^t\big) \leq \sup_{\mathcal{F}_{\text{ada}}(\varepsilon)} \sum_{t \in \mathcal{T}} I\big((\mathbf{P}_I^t(v), \mathbf{P}_V^t(v)); \mathbf{y}_v^t\big) \tag{22}$$

We next show that adaptive architectures can strictly outperform fixed architectures when there exist two timestamps, $t_1$ and $t_2$, for which the optimal architectures differ when tasked with discovering invariant patterns at $t_1$ and $t_2$, i.e., $\arg\max_\phi I(\mathbf{P}_I^{t_1}(v); \mathbf{y}_v^{t_1}) \neq \arg\max_\phi I(\mathbf{P}_I^{t_2}(v); \mathbf{y}_v^{t_2})$. Specifically, let $\phi_i$ and $\phi_j$ denote the optimal architectures at $t_1$ and $t_2$, respectively, with $i \neq j$. By the definition of optimality, we have:

$$\begin{aligned} \sup_{\phi_i} I(\mathbf{P}_I^{t_1}(v); \mathbf{y}_v^{t_1}) + \sup_{\phi_j} I(\mathbf{P}_I^{t_2}(v); \mathbf{y}_v^{t_2}) &> \sup_{\phi_i} I(\mathbf{P}_I^{t_1}(v); \mathbf{y}_v^{t_1}) + \sup_{\phi_i} I(\mathbf{P}_I^{t_2}(v); \mathbf{y}_v^{t_2}) \\ &> \sup_{\phi_j} I(\mathbf{P}_I^{t_1}(v); \mathbf{y}_v^{t_1}) + \sup_{\phi_j} I(\mathbf{P}_I^{t_2}(v); \mathbf{y}_v^{t_2}) \end{aligned} \tag{23}$$

The first inequality follows from the fact that $\phi_j$ outperforms $\phi_i$ at $t_2$, while the second follows because $\phi_i$ outperforms $\phi_j$ at $t_1$. Consequently, an adaptive architecture that applies $\phi_i$ at $t_1$ and $\phi_j$ at $t_2$ strictly outperforms any fixed architecture. For the remaining time steps, we can regard them as a subset $\mathcal{T} \subseteq \{1, \ldots, T\}$. By the result in Eq. 22, we therefore obtain:

$$\begin{aligned} \sup_{\mathcal{F}_{\text{ada}}(\varepsilon)} \sum_{t=1}^{T} I(\mathbf{P}_I^t(v); \mathbf{y}_v^t) &> \sup_{\phi_{\text{fix}}=\{\phi_i^t | t=1,2,\ldots,T\}} \sum_{t=1}^{T} I(\mathbf{P}_I^t(v); \mathbf{y}_v^t) \\ &> \sup_{\phi_{\text{fix}}=\{\phi_j^t | t=1,2,\ldots,T\}} \sum_{t=1}^{T} I(\mathbf{P}_I^t(v); \mathbf{y}_v^t) \end{aligned} \tag{24}$$

Combining Eq. 20 and Eq. 24, we obtain:

$$\begin{aligned} \sup_{\mathcal{F}_{\text{ada}}(\varepsilon)} \sum_{t=1}^{T} I\big((\mathbf{P}_I^t(v), \mathbf{P}_V^t(v)); \mathbf{y}_v^t\big) &> \sup_{\phi_{\text{fix}}=\{\phi_i^t | t=1,2,\ldots,T\}} \sum_{t=1}^{T} I\big((\mathbf{P}_I^t(v), \mathbf{P}_V^t(v)); \mathbf{y}_v^t\big) \\ &> \sup_{\phi_{\text{fix}}=\{\phi_j^t | t=1,2,\ldots,T\}} \sum_{t=1}^{T} I\big((\mathbf{P}_I^t(v), \mathbf{P}_V^t(v)); \mathbf{y}_v^t\big) \end{aligned} \tag{25}$$

In summary, adaptive encoder architectures are at least as effective as fixed architectures in capturing both invariant and variant patterns under the same invariance constraint. Moreover, they can strictly outperform fixed architectures when the optimal invariance-preserving encoder differs across timestamps, as this allows the model to adapt to varying distribution shifts, thereby capturing more total information and yielding invariant patterns and variant patterns more effectively.

## E  ADDITIONAL RELATED WORKS

**Out of Distribution Generalization.**  Most existing machine learning approaches rely on the assumption that training and test datasets are independently and identically distributed, an assumption often violated in practical scenarios (Arjovsky et al., 2019; Ahuja et al., 2020b; Shen et al., 2021; Lin et al., 2022; Bae et al., 2021; Li et al., 2024; Ge et al., 2025; Wang et al., 2025c; Tian et al.; Li et al., 2025c). In such cases, distribution shifts between training and test data can severely undermine model performance. To mitigate this, the study of Out-of-Distribution (OOD) generalization has gained substantial attention across a wide range of domains (Yao et al., 2022; Xu et al., 2024; Wang

et al., 2025d). Representative methods include Invariant Risk Minimization (IRM) (Arjovsky et al., 2019), which seeks predictors invariant across training domains by minimizing empirical risks jointly, thereby achieving consistent performance across diverse environments. GroupDRO (Sagawa et al., 2019) instead emphasizes robustness to worst-case groups by focusing optimization on domains with the highest error rates. Similarly, VREx (Krueger et al., 2021) reduces risk variance across domains, alleviating sensitivity to distributional changes. However, these methods fail to consider the unique challenges posed by graphs, such as complex relational structures and dependencies, which are crucial for effective OOD generalization in graph-based tasks.

**Graph Out of Distribution Generalization.** Graph out-of-distribution generalization must account for distribution shifts not only in node features but also in complex structural dependencies and relational patterns (Zhu et al., 2021; Fan et al., 2021; Li et al., 2022a; Chen et al., 2022; Qin et al., 2022; Gui et al., 2023; Chen et al., 2023b; Wu et al., 2023; Jia et al., 2024; Yao et al., 2024a; Chen et al., 2024; Yao et al., 2024b; Wang et al., 2025a; Li et al., 2026), where the challenges often stem from variations in topology, such as graph size or structural attributes. For example, Bevilacqua et al. (Bevilacqua et al., 2021) employ structural causal models under independence assumptions to learn representations transferable across different graph size distributions. G-mixup (Han et al., 2022) proposes a data augmentation strategy that interpolates node features and structures in embedding space to enhance robustness. Advances in graph self-supervised learning and graph explanation techniques further contribute to addressing structural distribution shifts (Li et al., 2022b). However, these methods often overlook the evolving nature of distribution shifts in dynamic graphs, which can significantly impact model performance over time.

