# OpenReview forum: "Adaptive Mixture of Disentangled Experts for Dynamic Graph Out-of-Distribution Generalization"
_ICLR.cc/2026/Conference — ICLR 2026 Poster_

### Official Review · Reviewer_h4tg · 2025-10-27

**Soundness:** 2
**Presentation:** 3
**Contribution:** 2
**Rating:** 4
**Confidence:** 5

**Summary:**

This paper addresses the challenge of dynamic graph learning under evolving distribution shifts. The authors propose AdaMix, an adaptive MoE framework that dynamically routes nodes to different GNN experts based on spatio-temporal distribution characteristics. The model includes a memory-augmented distribution detector, prototype-guided disentangled experts, and a distribution-aware intervention mechanism. Experiments on several real-world and synthetic dynamic graph datasets are reported to show improved OOD generalization compared to existing methods.

**Strengths:**

1. The proposed adaptive mixture formulation is interesting and aligns with the idea of conditional computation in dynamic systems.

2. Experimental results show consistent improvements over several baselines, including recent OOD graph learning models.

3. The inclusion of ablation studies and visualizations helps demonstrate the contribution of individual components.

**Weaknesses:**

1. Novelty is limited. The framework closely follows the structure of prior dynamic OOD works with mostly terminological changes (more like a combination of DIDA, SILD and EAGLE).

2. The “adaptive architecture” claim is somewhat overstated. The architecture is fixed after training, and only expert weights are adaptively combined during inference.

3. The motivation for modeling distribution shifts via expert routing, rather than latent environments or causal factors, is underdeveloped and lacks theoretical grounding.

4. The mathematical formulation remains descriptive rather than rigorous. The objective functions and routing updates are introduced heuristically without clear derivation.

5. The experiments do not provide sufficient diagnostic evaluation (e.g., temporal shift severity, expert specialization visualization) to validate the “adaptive” behavior claimed.

6. Writing style is occasionally redundant and derivative, with repeated phrases and limited conceptual clarity in the theoretical section.

**Questions:**

1. How is the proposed adaptive routing different in principle from a standard MoE gating mechanism conditioned on node embeddings?

2. During inference, does the model truly change its computation path, or merely reweight pre-trained experts?

3. What ensures that each expert captures a distinct distributional pattern rather than overlapping ones?

4. How sensitive is AdaMix to the number of experts? Were there cases where performance degraded with more experts?

5. Could the observed performance gains stem from increased model capacity rather than intrinsic adaptability to distribution shifts?

6. How does the proposed intervention mechanism differ from data augmentation or cross-domain mixing used in prior OOD graph works?

**Details Of Ethics Concerns:**

The manuscript exhibits unusually high structural and textual similarity to a previously published NeurIPS 2023 paper (*Environment-Aware Dynamic Graph Learning for Out-of-Distribution Generalization*), including similar methodological decomposition, variable notation, section layout, and figure organization, with only superficial terminology substitutions (e.g., replacing “environment” with “expert”). Although the earlier work is cited, the overlap appears to exceed standard academic referencing and may constitute **structural or paraphrased plagiarism**.

Moreover, the submission includes the statement:

> “To assist with language clarity and grammatical correctness, a large language model (LLM) was employed for proofreading and text refinement; however, all scientific content, ideas, analyses, and conclusions are solely the work of the authors.”

This disclaimer is ethically problematic. Given the high degree of overlap with prior literature, such phrasing could be interpreted as an attempt to attribute textual or conceptual similarities to LLM-based rewriting, thereby **deflecting author responsibility** and potentially misleading reviewers.

**Request:**
I respectfully ask the Ethics Committee:
1. Whether the submission reuses protected text, figures, or structural design from earlier work.
2. Whether the above LLM disclaimer constitutes an inappropriate use of AI attribution to obscure authorship accountability.

---

> ### Author Response · Authors · 2025-11-24
> **Author Response to Reviewer h4tg(1/3)**
>
> We would like to express our sincere gratitude to the reviewer for the detailed comments and insightful questions. We respond to each of the reviewer’s comments point by point as follows.
>
>
> > 1. "Novelty is limited. The framework closely follows the structure of prior dynamic OOD works with mostly terminological changes (more like a combination of DIDA, SILD and EAGLE)."
>
> Thank you for your thoughtful and valuable feedback. Prior works (e.g., DIDA, SILD, EAGLE) incorporate disentangling and intervention to extract invariant patterns for OOD generalization in dynamic graphs, and this forms the core idea and structure of these methods. In our study, we identify a unique challenge for dynamic-graph OOD—**evolving distribution shifts**—and further observe that existing approaches within this structure share a key limitation: fixed-architecture methods struggle to consistently remain optimal when tasked with discovering invariant patterns across different underlying distributions. Our method therefore aims to address this difficulty from an architectural perspective while adhering to the established pipeline for extracting invariant patterns, which naturally leads our framework to follow the overall structure used in prior dynamic OOD works. Specifically, we introduce two key modules: a spatio-temporal distribution detector and a prototype-guided mixture of disentangled experts, which together infer evolving distribution shifts and adaptively route disentangled experts to the corresponding shifted regimes. Moreover, inspired by the diverse expert selections that implicitly indicate different underlying distributions, we further develop a more efficient distribution-aware intervention mechanism that leverages expert choices to guide interventions across different distributions.
>
>
> > 2. "The “adaptive architecture” claim is somewhat overstated. The architecture is fixed after training, and only expert weights are adaptively combined during inference." and "During inference, does the model truly change its computation path, or merely reweight pre-trained experts?"
>
> Thank you for raising this important point. Our notion of adaptive architecture is defined relative to the underlying data distribution. In practice,different time periods or OOD scenarios induce distinct distributions, each of which may have an optimal architecture for extracting invariant patterns. We reweight the pre-trained experts according to the distribution inferred at inference time, thereby extracting the optimal invariant patterns for each distribution. We apologize for the confusion, and we have added clarifications in the contribution summary part (line 74) of the revised manuscript to more clearly articulate the intended meaning of “adaptive architecture”.
>
> > 3. "The motivation for modeling distribution shifts via expert routing, rather than latent environments or causal factors, is underdeveloped and lacks theoretical grounding."
>
> Thank you for raising this point. Some misunderstanding may exist: our method does not model distribution shifts through expert routing itself; rather, following the previous paradigm for extracting invariant patterns, we first infer the underlying distributional factors (which can be interpreted as the causal factors governing how distributions determine architectures) and then select the appropriate experts to extract invariant patterns. The overall motivation of our work is to address a common limitation of prior methods—namely, that fixed-architecture approaches often fail to consistently remain optimal when discovering invariant patterns across different underlying distributions (as discussed in Section 3). Therefore, it is necessary to infer the underlying distributional factors and dynamically select experts to extract the most effective invariant patterns.

---

> ### Author Response · Authors · 2025-11-24
> **Author Response to Reviewer h4tg(2/3)**
>
> > 4. "The mathematical formulation remains descriptive rather than rigorous. The objective functions and routing updates are introduced heuristically without clear derivation."
>
> We thank the reviewer for this feedback. While we do not provide full derivation steps, we clearly state the design motivations at the beginning of each module and describe the rationale and intended effects of each formulation following the corresponding equations, thereby clarifying the soundness of our design choices. For example, in Section 4.1, we first explain that to route experts adaptively under evolving distribution shifts, each expert must specialize in a distinct factor of variation in the data distribution so that expert routing aligns with the underlying distribution. Correspondingly, the disentanglement loss in Eq.2 is designed to enforce mutual dissimilarity among prototypes, thereby fostering disentanglement across experts and encouraging each expert to develop a distinct area of specialization. Similarly, the prototype-guided routing in Eq.3 assigns higher weights to experts whose prototypes are more similar to the current routing vector $\mathbf{r}^t$, directly supporting the intended alignment between inferred distributions and expert selection. Together, these formulations are consistent with and grounded in the design objectives stated earlier.
>
> > 5. "The experiments do not provide sufficient diagnostic evaluation (e.g., temporal shift severity, expert specialization visualization) to validate the “adaptive” behavior claimed."
>
> We thank the reviewer for the important suggestion. Appendix C.3 presents the temporal distribution shift severity across different datasets by showing how the number of nodes and their degrees vary over time. In addition, we have added Appendix C.6, which provides an illustrative showcase of the final architectures discovered for Nodes 0–2 in Aminer across different time steps, offering further validation of the claimed “adaptive” behavior.
>
> > 6. "Writing style is occasionally redundant and derivative, with repeated phrases and limited conceptual clarity in the theoretical section."
>
> Thank you for your valuable feedback. We acknowledge that some phrases were repeated and certain concepts were not presented as clearly as they could be. In the revision, we will streamline the exposition to remove redundant descriptions and reorganize the theoretical section to improve conceptual flow. These changes will result in a more concise and coherent presentation of the core ideas.
>
> > 7. "How is the proposed adaptive routing different in principle from a standard MoE gating mechanism conditioned on node embeddings?"
>
> We thank the reviewer for raising this point. Our approach differs from a standard Mixture of Experts (MoE) gating, where experts operate independently and do not explicitly model relational structures. First, standard MoE gating does not encourage that each expert corresponds to a meaningful or disentangled factor of variation. In contrast, in our setting—where experts must be adaptively routed under evolving distribution shifts—it is crucial for each expert to specialize in a distinct factor of variation. To this end, we associate each expert with a corresponding prototype. These prototypes are mutually disentangled, capture distinct factors of variation, and serve as anchors guiding the routing process. Second, standard MoE gating does not account for the dynamic nature of temporal graphs; by incorporating Memory-augmented Vectors, our model leverages historical information to better infer the current distribution, enabling more appropriate routing to the relevant experts.

---

> ### Author Response · Authors · 2025-11-24
> **Author Response to Reviewer h4tg(3/3)**
>
> > 8. "What ensures that each expert captures a distinct distributional pattern rather than overlapping ones?"
>
> Thank you for highlighting this important point. We associate each expert with a corresponding prototype, constructed using the disentanglement function defined in Eq.2. These prototypes are mutually disentangled, minimizing overlap between them. During training, the model parameters responsible for inferring the current distribution are updated jointly with the prototypes, which represent the factors captured by each expert. This ensures that each prototype captures a distinct factor of variation and serves as an anchor guiding the routing process.
>
> > 9. "How sensitive is AdaMix to the number of experts? Were there cases where performance degraded with more experts?"
>
> We thank the reviewer for the suggestion. To evaluate sensitivity to the number of experts, we added GraphConv to the original set of four expert architectures and examined the effect of increasing the number of experts. The results show that using five experts still maintains comparable performance.
>
> | Dataset        | Collab         | Yelp          | Link-syn (0.4) | Link-syn (0.6) | Link-syn (0.8) |
> |----------------|----------------|---------------|---------------|---------------|---------------|
> | Four experts | 84.85 ± 0.39   | 82.65 ± 0.87  | 90.21 ± 0.13  | 89.64 ± 0.26  | 88.86 ± 0.13  |
> | Five experts | 85.24 ± 0.27   | 83.59 ± 0.19  | 88.71 ± 0.41  | 89.34 ± 0.77  | 88.78 ± 0.42  |
>
> > 10. "Could the observed performance gains stem from increased model capacity rather than intrinsic adaptability to distribution shifts?"
>
> We thank the reviewer for raising this point. In our ablation study, we compared our adaptive routing strategy with a standard MoE gating strategy (**rep rou** in Section 5.3) while keeping the total number of parameters the same. The results show a significant performance drop for the standard gating, indicating that the gains are not simply due to increased model capacity. Additionally, we adapted a static-graph MoE method to our pipeline, and as shown in the table below, its performance remains substantially lower than ours. These results collectively demonstrate that the observed improvements are driven by our adaptive routing mechanism rather than the number of parameters.
>
> | Method      | Collab        | Yelp          | Aminer        | Link-syn (0.4) | Link-syn (0.6) | Link-syn (0.8) | Node-syn (0.4)       | Node-syn (0.6)       | Node-syn (0.8)       |
> |-------------|---------------|---------------|---------------|----------------|----------------|----------------|----------------|----------------|----------------|
> | GMoE    | 56.45 ± 0.56  | 72.53 ± 15.14 | 47.74 ± 1.05  | 55.39 ± 1.92   | 54.97 ± 4.94   | 56.30 ± 2.35   | 83.33 ± 1.04   | 80.83 ± 0.06   | 72.08 ± 1.31   |
> | GraphMetro | 57.92 ± 0.11 | 45.66 ± 10.59 | 48.15 ± 1.26 | 59.53 ± 0.08   | 59.28 ± 0.09   | 58.72 ± 0.12   | 75.82 ± 4.35   | 78.19 ± 3.53   | 75.25 ± 3.82   |
> | AdaMix (Ours)   | 84.85 ± 0.39  | 82.65 ± 0.87  | 51.34 ± 0.36  | 90.21 ± 0.13   | 89.64 ± 0.26   | 88.86 ± 0.13   | 83.63 ± 1.60   | 81.50 ± 0.38   | 76.19 ± 0.82   |
>
> > 11. "How does the proposed intervention mechanism differ from data augmentation or cross-domain mixing used in prior OOD graph works?"
>
> Thank you for raising this point. Traditional data augmentation aims to expand the sample distribution so that the model can adapt to more varied data, but this process is largely random and lacks explicit purpose. In contrast, our approach performs interventions based on variant patterns from different distributions, which is more targeted and can more efficiently improve the model’s generalization across samples with varying distributional patterns. Cross-domain mixing, on the other hand, assumes prior knowledge of different domains; however, in our setting, such information is not available. Our intervention mechanism leverages the disentangled experts and their corresponding prototypes to identify specific factors of variation associated with evolving distribution shifts. Since distribution differences influence expert selection, we can readily use the expert choices to guide interventions across different distributions. This allows interventions to be adaptive and factor-specific, rather than applying uniform intervention.
>
> **References to the methods mentioned in this response:**
>
> [1] DIDA: Dynamic Graph Neural Networks under Spatio-Temporal Distribution Shift
>
> [2] EAGLE: Environment-Aware Dynamic Graph Learning for Out-of-Distribution Generalization
>
> [3] SILD: Spectral Invariant Learning for Dynamic Graphs under Distribution Shifts
>
> [4] GraphMETRO: Mitigating Complex Graph Distribution Shifts via Mixture of Aligned Experts
>
> [5] GMoE: Graph Mixture of Experts: Learning on Large-Scale Graphs with Explicit Diversity Modeling

---

> ### Author Response · Authors · 2025-11-24
> **Response to Ethical Concerns**
>
> We thank the Reviewer for raising these points and would like to clarify the following to eliminate any misunderstandings:
>
> 	1. Alleged textual or structural similarities, including methodological decomposition, variable notation, and section layout:
> 		Our manuscript does not reuse or copy the structure, methodological decomposition, or text from the referenced NeurIPS 2023 paper (Environment-Aware Dynamic Graph Learning for Out-of-Distribution Generalization, method EAGLE). All sections, formulations, experiments, analyses, and conclusions are entirely original. From the perspective of methodological decomposition, our approach is fundamentally different from EAGLE. While EAGLE analyzes the limitations of previous dynamic OOD methods from an environment modeling perspective and decomposes the methodology through the “Modeling–Inferring–Discriminating–Generalizing” paradigm, our work focuses on limitations in dynamic-graph OOD from an architecture perspective. Specifically, we perform methodological decomposition based on the two key components of a mixture-of-experts (MoE) framework—the experts and the gating strategy—where the modules are responsible for extracting invariant patterns. Moreover, the reviewer’s comment suggesting “only superficial terminology substitutions (e.g., replacing ‘environment’ with ‘expert’)” reflects a misunderstanding. In EAGLE, the concept of an “environment” refers to certain factors that assist invariant patterns in predicting labels, whereas in our work, the experts constitute the model architecture specifically designed to extract better invariant patterns in our method. These two concepts operate at fundamentally different levels. Some perceived similarity in notions or terminology may be limited to general scientific concepts commonly used in dynamic-graph OOD studies, as seen in prior works such as DIDA, EAGLE, and SILD.
> 	2. On figure organization:
> 		We followed the style of EAGLE’s clear and intuitive layout for ablation and hyperparameter sensitivity plots, which may cause our figures to appear visually similar to prior work. However, the underlying experiments, data, and results are entirely original. Following the figure style of a prior paper, which does not affect the methodological design, experimental results, or conclusions, is solely intended to present the results more clearly and intuitively. To clarify this further, we will add explicit citations in figure captions where the style was followed, and we are willing to adjust figure style to further distinguish our figures, or to avoid potential issues if the figure style is protected.
> 	3. On LLM usage statement:
> 		We used a large language model solely for language clarity and grammatical correctness. We take full responsibility for all content, experiments, and conclusions, and we do not attribute any textual or conceptual similarities to LLM-based rewriting.
>
> We sincerely apologize for any confusion and reaffirm that all scientific contributions, experiments, and analyses in our manuscript are fully original, for which we take full responsibility.

---

> ### Comment · Reviewer_h4tg · 2025-11-25
>
> Thank the authors for their detailed responses, and great efforts in mitigating the misunderstandings, especially on the ethical issues. Most of my technical concerns have been addressed, and I suggest the authors carefully revise the manuscripts to include all the promised improvements. Overall, an interesting perspective on graph OOD, I would like to recommend its acceptance.

---

> > ### Author Response · Authors · 2025-11-28
> > **Official Comment by Authors**
> >
> > We sincerely appreciate your insightful suggestion once again, which has been invaluable in refining our work. We have revised the manuscript to incorporate all the promised improvements and would be grateful for any further comments or concerns you may have.

---

### Official Review · Reviewer_1zib · 2025-10-30

**Soundness:** 3
**Presentation:** 2
**Contribution:** 2
**Rating:** 6
**Confidence:** 4

**Summary:**

This paper investigates dynamic graph representation learning under evolving distribution shifts and introduces an adaptive-architecture framework, AdaMix, designed to handle such changes over time. The proposed approach employs a mixture of architecture experts, guided by a spatio-temporal distribution detector, a prototype-based expert routing mechanism, and a distribution-aware intervention module to capture invariant spatio-temporal patterns. The authors identify and address three main challenges: detecting evolving shifts, dynamically routing experts, and ensuring effective invariant learning. Experimental results on synthetic and real-world datasets are reported to demonstrate the method’s performance compared with existing approaches.

**Strengths:**

1. The problem studied in this paper is important.

2. The experimental datasets are sufficient.

3. The work has a certain degree of theoretical support.

**Weaknesses:**

1. Some techniques are applied too directly without sufficient explanation. For example, why is invariant pattern modeling performed in the spectral domain?

2. The comparison methods in the experiments are not up to date (latest from 2023), lacking evaluations against GraphMoE-type approaches such as GMoE and GraphMETRO.

3. Considering that the proposed method follows an ensemble learning paradigm and incorporates more GNN encoders than the baselines, the performance gains on real-world datasets are relatively small—mostly within a 1% range. This raises the question of whether such limited improvement justifies the increased model complexity.

4. The method performs better on synthetic datasets but only moderately on real-world ones. This discrepancy suggests that the paper’s assumptions might be overly idealized and not well aligned with the characteristics of real-world data distributions.

**Questions:**

Please see the weaknesses.

---

> ### Author Response · Authors · 2025-11-24
> **Author Response to Reviewer 1zib(1/2)**
>
> We would like to express our sincere appreciation to the reviewer for providing us with detailed suggestions. We have carefully reviewed each comment and offer the following responses.
>
> > 1. "Some techniques are applied too directly without sufficient explanation. For example, why is invariant pattern modeling performed in the spectral domain?"
>
> Thank you for bringing this to our attention. Inspired by prior work (SILD), which shows that certain distribution shifts may be unobservable in the time domain but become apparent in the spectral domain, we conduct invariant pattern modeling in the spectral domain to incorporate this spectral information. We have provided more detailed explanations in this section in the revised manuscript(line 289).
>
> > 2. "The comparison methods in the experiments are not up to date (latest from 2023), lacking evaluations against GraphMoE-type approaches such as GMoE and GraphMETRO."
>
> Thank you for raising this point. Since prior GraphMoE methods were primarily designed for static graph tasks, we treat each snapshot of the discrete dynamic graph as an individual static graph and adapt both GMoE and GraphMETRO accordingly. For a fair comparison, we retain their invariant and variant pattern extraction components and align with the OOD evaluation setting. The experimental results are shown below. We observe that GMoE and GraphMETRO do not achieve competitive performance, likely because their architectures do not model the temporal dependencies across different time steps in dynamic graphs.
>
> | Method      | Collab        | Yelp          | Aminer        | Link-syn (0.4) | Link-syn (0.6) | Link-syn (0.8) | Node-syn (0.4)       | Node-syn (0.6)       | Node-syn (0.8)       |
> |-------------|---------------|---------------|---------------|----------------|----------------|----------------|----------------|----------------|----------------|
> | GMoE    | 56.45 ± 0.56  | 72.53 ± 15.14 | 47.74 ± 1.05  | 55.39 ± 1.92   | 54.97 ± 4.94   | 56.30 ± 2.35   | 83.33 ± 1.04   | 80.83 ± 0.06   | 72.08 ± 1.31   |
> | GraphMetro | 57.92 ± 0.11 | 45.66 ± 10.59 | 48.15 ± 1.26 | 59.53 ± 0.08   | 59.28 ± 0.09   | 58.72 ± 0.12   | 75.82 ± 4.35   | 78.19 ± 3.53   | 75.25 ± 3.82   |
> | AdaMix (Ours)   | 84.85 ± 0.39  | 82.65 ± 0.87  | 51.34 ± 0.36  | 90.21 ± 0.13   | 89.64 ± 0.26   | 88.86 ± 0.13   | 83.63 ± 1.60   | 81.50 ± 0.38   | 76.19 ± 0.82   |

---

> ### Author Response · Authors · 2025-11-24
> **Author Response to Reviewer 1zib(2/2)**
>
> > 3. "Considering that the proposed method follows an ensemble learning paradigm and incorporates more GNN encoders than the baselines, the performance gains on real-world datasets are relatively small—mostly within a 1% range. This raises the question of whether such limited improvement justifies the increased model complexity."
>
> Thank you for bringing this to our attention. Although AdaMix improves over the second-best baseline by less than 1% on each individual real-world dataset, a closer look shows that these baselines are far from consistently strong across datasets. For example, the second-best performer on Yelp, SILD-GIN, performs poorly on both Collab and Aminer—on Collab, it is 9% lower than AdaMix. Conversely, the second-best baseline on Collab, EAGLE, drops by 5% compared to AdaMix on Yelp. This inconsistency highlights exactly the limitation we aim to address: fixed-architecture methods struggle to consistently remain optimal when tasked with discovering invariant patterns across different underlying distributions. When we compute the average performance across all three real datasets, AdaMix delivers a clear improvement of more than 1.8% over the second-best baseline, SILD, with only a modest increase in time cost(shown in the second table). This demonstrates its effectiveness and its reduced sensitivity to distributional variation.
>
> | Method        | Collab | Yelp  | Aminer  | Average performance |
> |---------------|--------|-------|----------------------|----------------------|
> | DIDA          | 81.87  | 75.92 | 48.82               | 68.87               |
> | EAGLE         | 84.41  | 77.26 | 50.77               | 70.81               |
> | SILD          | 84.09  | 78.65 | 50.67               | 71.14               |
> | SILD-GCN      | 79.53  | 43.74 | 48.55               | 57.27               |
> | SILD-GAT      | 83.82  | 50.18 | 50.16               | 61.39               |
> | SILD-GIN      | 75.18  | 81.55 | 41.29               | 66.01               |
> | SILD-GATv2    | 83.97  | 47.84 | 50.07               | 60.63               |
> | AdaMix (Ours) | 84.85  | 82.65 | 51.34               | 72.95               |
>
>
> | Dataset| EAGLE Train | EAGLE Inf | SILD Train | SILD Inf | AdaMix Train | AdaMix Inf |
> |----------------|-------------|-----------|------------|----------|--------------|------------|
> | Yelp           | 6.84        | 0.87      | 0.93       | 0.74     | 0.78         | 0.67       |
> | Collab         | 14.160      | 3.884     | 0.37       | 0.20     | 0.45         | 0.56       |
> | Link-syn (0.4) | 4.96        | 1.23      | 0.35       | 0.36     | 0.47         | 0.71       |
> | Link-syn (0.6) | 7.77        | 1.99      | 0.27       | 0.31     | 0.32         | 0.62       |
> | Link-syn (0.8) | 11.02       | 2.20      | 0.53       | 0.34     | 0.48         | 0.69       |
> | Aminer         | 9.22        | 0.18      | 0.31       | 0.27     | 1.03         | 1.15       |
> | Node-syn (0.4) | 3.44        | 0.43      | 0.29       | 0.16     | 1.04         | 0.55       |
> | Node-syn (0.6) | 3.47        | 0.44      | 0.23       | 0.12     | 1.17         | 0.60       |
> | Node-syn (0.8) | 3.48        | 0.44      | 0.22       | 0.12     | 1.14         | 0.60       |
>
> > 4. "The method performs better on synthetic datasets but only moderately on real-world ones. This discrepancy suggests that the paper’s assumptions might be overly idealized and not well aligned with the characteristics of real-world data distributions."
>
> Thank you for raising this important point. You are correct that our method exhibits larger performance gains on synthetic datasets and more moderate improvements on real-world datasets. This gap is expected, as synthetic datasets are constructed to align more closely with the assumptions made in our modeling framework, allowing the underlying mechanisms to be validated in a controlled setting.
> However, as discussed earlier, our method still delivers a noticeable improvement when averaging results across the three real-world datasets. The clearer gains observed on the synthetic datasets further strengthen the evidence for the soundness of our design, showing that when the assumed conditions hold more precisely, the method’s advantages become even more pronounced.
>
> **References to the methods mentioned in this response:**
>
> [1] DIDA: Dynamic Graph Neural Networks under Spatio-Temporal Distribution Shift
>
> [2] EAGLE: Environment-Aware Dynamic Graph Learning for Out-of-Distribution Generalization
>
> [3] SILD: Spectral Invariant Learning for Dynamic Graphs under Distribution Shifts
>
> [4] GraphMETRO: Mitigating Complex Graph Distribution Shifts via Mixture of Aligned Experts
>
> [5] GMoE: Graph Mixture of Experts: Learning on Large-Scale Graphs with Explicit Diversity Modeling

---

> > ### Comment · Reviewer_1zib · 2025-11-28
> >
> > Thanks for the author's reply, which has solved my confusion. The added experiments have enhanced the credibility of the method. Indeed, using different expert models to deal with complex distribution shifts is beneficial and interesting. Therefore, I will further raise the score.

---

> > > ### Author Response · Authors · 2025-11-30
> > > **Official Comment by Authors**
> > >
> > > We sincerely thank you once again for your thorough review and constructive feedback on our submission.

---

### Official Review · Reviewer_fPRk · 2025-10-31

**Soundness:** 2
**Presentation:** 3
**Contribution:** 2
**Rating:** 4
**Confidence:** 4

**Summary:**

This paper aims to address the challenge of dynamic graph representation learning under evolving distribution shifts, a limitation of existing fixed-architecture methods. It proposes an adaptive-architecture framework for this task, comprising three core components: a spatio-temporal distribution detector, prototype-guided disentangled experts, and a distribution-aware intervention mechanism. Extensive experiments show the proposed model outperforms state-of-the-art baselines (e.g., SILD, EAGLE) in link prediction and node classification.

**Strengths:**

1. Using MoE technology to solve the problem of distribution shifts on dynamic graphs is interesting and effective.

2. The distribution-aware mechanism avoids inefficient random interventions by leveraging dominant experts, enhancing invariant pattern extraction.

3. Experiment results on diverse synthetic datasets are good.

**Weaknesses:**

1. Apart from the MoE part, there is existing work on both disentangling (e.g., DIDA, SILD) and intervention (e.g., SILD, EAGLE) on dynamic graphs. Therefore, the overall architecture has a piecemeal feel. It is impossible to discern any fundamental changes in these sections compared to existing work.

2. The MoE section (section 4.1) has relatively low innovation.

3. The performance improvement on real datasets is very limited (less than 1%).

**Questions:**

Please refer to the weakness.

---

> ### Author Response · Authors · 2025-11-24
> **Author Response to Reviewer fPRk(1/2)**
>
> We sincerely thank the reviewer for their valuable feedback. We have carefully considered each point and address them as follows.
>
> > 1. "Apart from the MoE part, there is existing work on both disentangling (e.g., DIDA, SILD) and intervention (e.g., SILD, EAGLE) on dynamic graphs. Therefore, the overall architecture has a piecemeal feel. It is impossible to discern any fundamental changes in these sections compared to existing work."
>
> We appreciate your valuable feedback. Prior works (e.g., DIDA, SILD, EAGLE) indeed rely on disentangling and intervention to extract invariant patterns for OOD generalization in dynamic graphs, which forms the core idea and pipeline of these methods. In our study, we identify a unique challenge for dynamic-graph OOD-**evolving distribution shifts**—and further observe that existing approaches within this pipeline share a key limitation: fixed-architecture methods struggle to consistently remain optimal when tasked with discovering invariant patterns across different underlying distributions. Our method therefore aims to address this difficulty from an architectural perspective while adhering to the established pipeline for extracting invariant patterns, which naturally leads our framework to follow the overall pipeline used in prior dynamic OOD works. Specifically, we first introduce two key modules: a spatio-temporal distribution detector and a prototype-guided mixture of disentangled experts, which together infer evolving distribution shifts and adaptively route disentangled experts to the corresponding shifted regimes. Moreover, inspired by the diverse expert selections that implicitly indicate different underlying distributions, we further develop a more efficient distribution-aware intervention mechanism.
>
> > 2. "The MoE section (section 4.1) has relatively low innovation."
>
> Thank you for raising this point. The MoE section (Section 4.1) introduces the prototype-guided mixture of disentangled experts module, which fundamentally differs from a standard Mixture of Experts, where experts operate independently and lack explicit relational modeling. Such conventional architectures do not encourage that each expert corresponds to a meaningful or disentangled factor of variation. In our setting, however—where experts must be adaptively routed under **evolving distribution shifts**—it is crucial that each expert specializes in a distinct factor of variation to remain aligned with the underlying distributions. To achieve this, we associate each expert with a corresponding prototype. These prototypes are mutually disentangled, capture distinct factors of variation, and serve as anchors that guide the routing process. In the revised manuscript, we have emphasized our motivation at the beginning of this module and contrast it with standard MoE framework to highlight the innovation of our proposed module(line 202).

---

> ### Author Response · Authors · 2025-11-24
> **Author Response to Reviewer fPRk(2/2)**
>
> > 3. "The performance improvement on real datasets is very limited (less than 1%)."
>
> Thank you for bringing this to our attention. Although AdaMix improves over the second-best baseline by less than 1% on each individual real-world dataset, a closer look shows that these baselines are far from consistently strong across datasets. For example, the second-best performer on Yelp, SILD-GIN, performs poorly on both Collab and Aminer—on Collab, it is 9% lower than AdaMix. Conversely, the second-best baseline on Collab, EAGLE, drops by 5% compared to AdaMix on Yelp. This inconsistency highlights exactly the limitation we aim to address: fixed-architecture methods struggle to consistently remain optimal when tasked with discovering invariant patterns across different underlying distributions. When we compute the average performance across all three real datasets, AdaMix delivers a clear improvement of more than 1.8% over the second-best baseline, SILD, with only a modest increase in time cost(shown in the second table). This demonstrates its effectiveness and its reduced sensitivity to distributional variation.
>
> | Method        | Collab | Yelp  | Aminer  | Average performance |
> |---------------|--------|-------|----------------------|----------------------|
> | DIDA          | 81.87  | 75.92 | 48.82               | 68.87               |
> | EAGLE         | 84.41  | 77.26 | 50.77               | 70.81               |
> | SILD          | 84.09  | 78.65 | 50.67               | 71.14               |
> | SILD-GCN      | 79.53  | 43.74 | 48.55               | 57.27               |
> | SILD-GAT      | 83.82  | 50.18 | 50.16               | 61.39               |
> | SILD-GIN      | 75.18  | 81.55 | 41.29               | 66.01               |
> | SILD-GATv2    | 83.97  | 47.84 | 50.07               | 60.63               |
> | AdaMix (Ours) | 84.85  | 82.65 | 51.34               | 72.95               |
>
>
> | Dataset        | EAGLE Train | EAGLE Inf | SILD Train | SILD Inf | AdaMix Train | AdaMix Inf |
> |----------------|-------------|-----------|------------|----------|--------------|------------|
> | Yelp           | 6.84        | 0.87      | 0.93       | 0.74     | 0.78         | 0.67       |
> | Collab         | 14.160      | 3.884     | 0.37       | 0.20     | 0.45         | 0.56       |
> | Link-syn (0.4) | 4.96        | 1.23      | 0.35       | 0.36     | 0.47         | 0.71       |
> | Link-syn (0.6) | 7.77        | 1.99      | 0.27       | 0.31     | 0.32         | 0.62       |
> | Link-syn (0.8) | 11.02       | 2.20      | 0.53       | 0.34     | 0.48         | 0.69       |
> | Aminer         | 9.22        | 0.18      | 0.31       | 0.27     | 1.03         | 1.15       |
> | Node-syn (0.4) | 3.44        | 0.43      | 0.29       | 0.16     | 1.04         | 0.55       |
> | Node-syn (0.6) | 3.47        | 0.44      | 0.23       | 0.12     | 1.17         | 0.60       |
> | Node-syn (0.8) | 3.48        | 0.44      | 0.22       | 0.12     | 1.14         | 0.60       |
>
>
> **References to the methods mentioned in this response:**
>
> [1] DIDA: Dynamic Graph Neural Networks under Spatio-Temporal Distribution Shift
>
> [2] EAGLE: Environment-Aware Dynamic Graph Learning for Out-of-Distribution Generalization
>
> [3] SILD: Spectral Invariant Learning for Dynamic Graphs under Distribution Shifts

---

> > ### Comment · Reviewer_fPRk · 2025-11-26
> >
> > Thanks to the authors for their efforts and responses. I believe most of my concerns have been addressed and will raise the score.

---

> > > ### Author Response · Authors · 2025-11-28
> > > **Official Comment by Authors**
> > >
> > > We sincerely appreciate your insightful suggestion once again, which has greatly helped us refine our work. Please do not hesitate to share any further comments or concerns.

---

### Official Review · Reviewer_CVZa · 2025-11-01

**Soundness:** 3
**Presentation:** 2
**Contribution:** 2
**Rating:** 4
**Confidence:** 3

**Summary:**

This paper addresses dynamic graph representation learning under evolving distribution shifts, a setting where both graph topology and features change over time and the nature of distribution shifts itself evolves. The authors propose AdaMix, an adaptive mixture-of-experts (MoE) framework that dynamically adjusts model architectures to capture invariant patterns across time. Extensive experiments on real-world (Collab, Yelp, Aminer) and synthetic datasets show that AdaMix consistently outperforms both standard dynamic GNNs (e.g., DySAT, EGCN) and prior OOD methods (DIDA, EAGLE, SILD).

**Strengths:**

1. The paper argues that evolving shifts make fixed architectures sub-optimal, which is well-supported.
2. The paper proposes a three component framework to dynamically adjust the architecture

**Weaknesses:**

1. Some baselines show very high variance (e.g., Aminer with SILD has huge std), while AdaMix gains on real data are sometimes modest. For examples, on Aminer15, Aminer16 and Aminer17, Adamix does not have evident improvements. The reported mean ± std of baselines often overlap or even exceed AdaMix in mean.
2. Adaptive MoE at node-time granularity plus FFT-domain masking and memory updates may increase training/inference cost.
3. Although an ablation study is reported, the performance drop after removing individual components (e.g., memory module, prototype disentanglement, distribution-aware intervention) is not quantitatively large or clearly demonstrated in the main paper. The results do not convincingly show that each component is essential to the final performance.

**Questions:**

1. What are the training/inference time compared with baselines?
2. Could you add a “no-FFT” variant (time-domain only) and a “no-memory but deeper router” variant to disentangle spectral vs. temporal-memory contributions.

---

> ### Author Response · Authors · 2025-11-24
> **Author Response to Reviewer CVZa(1/2)**
>
> We sincerely thank the reviewer for their detailed comments and insightful questions. We have carefully considered all feedback and provide our responses to each point below.
>
> > 1. "Some baselines show very high variance (e.g., Aminer with SILD has huge std), while AdaMix gains on real data are sometimes modest. For examples, on Aminer15, Aminer16 and Aminer17, Adamix does not have evident improvements. The reported mean ± std of baselines often overlap or even exceed AdaMix in mean."
>
> Thank you for raising this important point. You are correct that some baselines exhibit high variance, particularly on real-world datasets such as Aminer, where methods like SILD show large standard deviations and may even overlap with or exceed AdaMix in mean. However, the second-best baselines do not perform consistently across all datasets. For example, SILD-GIN performs well on Yelp but drops significantly on Collab and Aminer, while EAGLE performs well on Aminer but performs substantially on Yelp. This inconsistency highlights exactly the limitation we aim to address: fixed-architecture methods struggle to consistently remain optimal when tasked with discovering invariant patterns across different underlying distributions. When averaging over the three real-world datasets, AdaMix achieves a clear improvement (>1.8%) over the second-best baseline, with only a modest increase in time cost. This demonstrates its effectiveness and its reduced sensitivity to distributional variation. Moreover, AdaMix often shows more stable performance, reflected by small standard deviations, indicating enhanced robustness to distribution shifts.
>
> | Method        | Collab | Yelp  | Aminer  | Average performance |
> |---------------|--------|-------|----------------------|----------------------|
> | DIDA          | 81.87  | 75.92 | 48.82               | 68.87               |
> | EAGLE         | 84.41  | 77.26 | 50.77               | 70.81               |
> | SILD          | 84.09  | 78.65 | 50.67               | 71.14               |
> | SILD-GCN      | 79.53  | 43.74 | 48.55               | 57.27               |
> | SILD-GAT      | 83.82  | 50.18 | 50.16               | 61.39               |
> | SILD-GIN      | 75.18  | 81.55 | 41.29               | 66.01               |
> | SILD-GATv2    | 83.97  | 47.84 | 50.07               | 60.63               |
> | AdaMix (Ours) | 84.85  | 82.65 | 51.34               | 72.95               |
>
>
> > 2. "Adaptive MoE at node-time granularity plus FFT-domain masking and memory updates may increase training/inference cost." and "What are the training/inference time compared with baselines?"
>
> Thank you for bringing up this point. Our method does introduce some additional time cost, but the increase is within an acceptable range. Following your suggestion, we evaluated the training and inference cost of our approach compared with competitive baselines. The training time is measured on the portion of the “W/O DS” dataset that requires loss computation and backpropagation (i.e., its training split). The inference time is measured on the remaining parts of the dataset, including the validation and test splits of “W/O DS” as well as the entire “W DS” dataset. The results are as follows. The table reports, for each method and each dataset, the average per-epoch training and inference cost (in seconds). All measurements are obtained under the same hardware configuration for a fair comparison:
>
> | Dataset        | EAGLE Train | EAGLE Inf | SILD Train | SILD Inf | AdaMix Train | AdaMix Inf |
> |----------------|-------------|-----------|------------|----------|--------------|------------|
> | Yelp           | 6.84        | 0.87      | 0.93       | 0.74     | 0.78         | 0.67       |
> | Collab         | 14.160      | 3.884     | 0.37       | 0.20     | 0.45         | 0.56       |
> | Link-syn (0.4) | 4.96        | 1.23      | 0.35       | 0.36     | 0.47         | 0.71       |
> | Link-syn (0.6) | 7.77        | 1.99      | 0.27       | 0.31     | 0.32         | 0.62       |
> | Link-syn (0.8) | 11.02       | 2.20      | 0.53       | 0.34     | 0.48         | 0.69       |
> | Aminer         | 9.22        | 0.18      | 0.31       | 0.27     | 1.03         | 1.15       |
> | Node-syn (0.4) | 3.44        | 0.43      | 0.29       | 0.16     | 1.04         | 0.55       |
> | Node-syn (0.6) | 3.47        | 0.44      | 0.23       | 0.12     | 1.17         | 0.60       |
> | Node-syn (0.8) | 3.48        | 0.44      | 0.22       | 0.12     | 1.14         | 0.60       |
>
> We can observe that environment-modeling methods such as EAGLE incur substantially higher time cost compared to non-environment-modeling approaches. In contrast, our AdaMix introduces only modest overhead relative to SILD.

---

> ### Author Response · Authors · 2025-11-24
> **Author Response to Reviewer CVZa(2/2)**
>
> > 3."Although an ablation study is reported, the performance drop after removing individual components (e.g., memory module, prototype disentanglement, distribution-aware intervention) is not quantitatively large or clearly demonstrated in the main paper. The results do not convincingly show that each component is essential to the final performance." and "Could you add a “no-FFT” variant (time-domain only) and a “no-memory but deeper router” variant to disentangle spectral vs. temporal-memory contributions."
>
> Thank you for your suggestion. Although removing individual components does not lead to a large performance drop on all datasets, some datasets exhibit clear degradation. For example, removing the memory module results in noticeable declines on Yelp, Link-syn (0.4), and Link-syn (0.6), and also causes the standard deviation to increase substantially. This indicates that the model becomes less stable without this module, highlighting its importance. Following your suggestion, we additionally include a “no-FFT” variant (time-domain only) and a “no-memory but deeper router” variant. The results are shown below:
>
> | Method                        | Collab         | Yelp           | Aminer        | Link-syn (0.4) | Link-syn (0.6) | Link-syn (0.8) |
> |-------------------------------|----------------|----------------|---------------|----------------|----------------|----------------|
> | FULL                          | 84.85 ± 0.39   | 82.65 ± 0.87   | 51.34 ± 0.36  | 90.21 ± 0.13   | 89.64 ± 0.26   | 88.86 ± 0.13   |
> | no-FFT                        | 61.19 ± 3.95   | 81.17 ± 9.34   | 46.96 ± 0.99  | 75.05 ± 3.59   | 81.97 ± 3.14   | 80.48 ± 5.64   |
> | no-memory but deeper router    | 67.70 ± 6.72   | 65.81 ± 3.06   | 50.44 ± 1.55  | 60.65 ± 0.63   | 65.01 ± 1.20   | 61.56 ± 2.29   |
>
> Based on the experimental results, we can draw the following conclusions: i) The “no-FFT” variant, which relies solely on time-domain information, shows noticeable declines compared to the full model, particularly on Collab. This demonstrates that spectral-domain invariant pattern modeling effectively captures distribution shifts that may be unobservable in the time domain but become evident in the spectral domain. ii) The “no-memory but deeper router” variant results in a noticeable performance drop, suggesting that the historical distribution information stored in memory vectors enables better inference of the current distribution, while disentangled prototypes allow the router to more effectively distinguish between different distributions.
>
>
> **References to the methods mentioned in this response:**
>
> [1] DIDA: Dynamic Graph Neural Networks under Spatio-Temporal Distribution Shift
>
> [2] EAGLE: Environment-Aware Dynamic Graph Learning for Out-of-Distribution Generalization
>
> [3] SILD: Spectral Invariant Learning for Dynamic Graphs under Distribution Shifts

---

### Author Response · Authors · 2025-11-28
**General Response by Authors**

Dear Reviewers, ACs, and PCs,

We would like to thank the ACs and PCs for handling our paper and extend our sincere gratitude to the four reviewers for their insightful and constructive feedback. We have addressed the comments in the individual responses and updated the paper accordingly, with key changes highlighted in red. With the discussion now concluded, we summarize our main contributions and our main revisions to the reviewers’ concerns as follows.

---
**Contributions**

- **Architecture Perspective for Dynamic Graph OOD:** We discover a unique challenge for dynamic graph OOD scenarios, namely evolving distribution shifts, and further observe that existing approaches share a key limitation: fixed-architecture methods struggle to remain consistently optimal when tasked with discovering invariant patterns across different underlying distributions. `reviewer CVZa, 1zib`

- **Novel Methodology for Adaptive Architectures:**  We propose an adaptive-architecture design framework, named AdaMix, to handle evolving distribution shifts in dynamic graphs. Specifically, it employs an MoE-based framework that accounts for both temporal dynamics and distributional changes, enabling the architecture to adapt to the underlying data distribution and discover invariant patterns. `reviewer CVZa, fPRk`

- **Extensive and Favorable Experimental Results:** Our method shows consistent improvements over several baselines. It also outperforms recent dynamic-graph OOD learning approaches across a wide range of datasets, particularly on the synthetic ones. `reviewer fPRk, 1zib, h4tg`


---

**Main Revisions**

- **For reviewer 1zib:** In Section 5.2, we add two static-graph MoE methods and evaluate them on real-world datasets(Table 1) and synthetic datasets(Table 2).

- **For reviewer CVZa:** In Section 5.3, we add two ablated versions, ‘w/o FFT’ and ‘w/o mem&rou’, to further evaluate the contributions of each module.

- **For reviewer h4tg:** In Appendix C.6, we add a showcase to illustrate the adaptive architecture.

- **For reviewer CVZa:** In Appendix C.7, we evaluate the training and inference costs of our approach compared with competitive baselines.

- **For reviewer h4tg:** In Appendix C.8, we test our method with more experts to evaluate its sensitivity to the number of experts.

- **For reviewer h4tg:** In Appendix E, we enhance the clarity of our theoretical derivation process.

- **For reviewer h4tg, fPRk and 1zib:** Other minor revisions to clarify the concepts (line 74) and motivation (lines 202, 289) presented in our paper.

---

Finally, we would once again like to express our sincere gratitude for your contributions to the conference.

Sincerely,

ICLR 2026 Submission 982 Authors

---

### Meta-Review · Area_Chair_Li5o · 2025-12-23

**Summary:**

This paper proposes a novel adaptive-architecture framework for dynamic graphs under evolving distribution shifts. Experimental results highlight a key limitation of existing approaches: fixed-architecture models struggle to remain consistently optimal when identifying invariant patterns in the presence of evolving distribution shifts. The paper further strengthens this finding by providing theoretical analysis that formally characterizes the limitations of fixed architectures in such settings. Overall, the proposed framework is well motivated and aligns naturally with the principles of adaptive architectures and the characteristics of dynamic distribution shifts. Extensive experiments demonstrate consistent performance improvements over a range of strong baselines, with particularly clear advantages on synthetic datasets.

Reviewers initially questioned the component necessity, and empirical grounding of the work. The authors addressed these concerns by supplementing extensive experiments (e.g., new baselines and ablations), enhancing theoretical clarity, and providing runtime and sensitivity analyses, which have addressed most of the reviewers’ concerns.

Overall, I think this is a good paper with significant contributions that deserve attention from the community.

**Reviewer Concerns:**

The reviewers mainly raised questions regarding the novelty of the paper, the insignificant improvement in the experimental results compared to the baseline, and the insufficiency of the ablation experiment. In terms of the experiment, I think the content supplemented by the authors in the rebuttal stage is convincing and indicates the key to the improvement of the method. In terms of novelty, the author's response is strong enough. Although the reviewers did not reply, they might agree with the authors' statement.

**Reviewer Scores:**

For Reviewer CVZa, I think the author's response is excellent and the reviewer is very likely to increase the score.
For Reviewer h4tg and fPRk, the reviewers have expressed their willingness to increase the score.
For Reviewer 1zib, the reviewer expressed a positive acceptance at the very beginning.
Overall, I think the overall evaluation of this paper will be positive in the end, so I am willing to accept it.

---

### Decision · Program_Chairs · 2026-01-26

Accept (Poster)